# A coupling prescription for post-Newtonian corrections in Quantum Mechanics

**Jelle Hartong,**[1] **Emil Have,**[1] **Niels A. Obers,**[2,3] **Igor Pikovski**[4,5]

[1] *School of Mathematics and Maxwell Institute for Mathematical Sciences,*
*University of Edinburgh, Peter Guthrie Tait Road, Edinburgh EH9 3FD, UK*

[2] *Nordita, KTH Royal Institute of Technology and Stockholm University,*
*Roslagstullsbacken 23, SE-106 91 Stockholm, Sweden*

[3] *The Niels Bohr Institute, University of Copenhagen*
*Blegdamsvej 17, DK-2100 Copenhagen Ø, Denmark*

[4] *Department of Physics, Stevens Institute of Technology*
*Castle Point on the Hudson, Hoboken, NJ 07030, USA*

[5] *Department of Physics, Stockholm University*
*AlbaNova University Center, SE-106 91 Stockholm, Sweden*

*E-mail:* j.hartong@ed.ac.uk, emil.have@ed.ac.uk, obers@nbi.dk,
igor.pikovski@fysik.su.se

ABSTRACT: The interplay between quantum theory and general relativity remains one of the main challenges of modern physics. A renewed interest in the low-energy limit is driven by the prospect of new experiments that could probe this interface. Here we develop a covariant framework for expressing post-Newtonian corrections to Schrödinger's equation on arbitrary gravitational backgrounds based on a $1/c^2$ expansion of Lorentzian geometry, where $c$ is the speed of light. Our framework provides a generic coupling prescription of quantum systems to gravity that is valid in the intermediate regime between Newtonian gravity and General Relativity, and that retains the focus on geometry. At each order in $1/c^2$ this produces a nonrelativistic geometry to which quantum systems at that order couple. By considering the gauge symmetries of both the nonrelativistic geometries and the $1/c^2$ expansion of the complex Klein–Gordon field, we devise a prescription that allows us to derive the Schrödinger equation and its post-Newtonian corrections on a gravitational background order-by-order in $1/c^2$. We also demonstrate that these results can be obtained from a $1/c^2$ expansion of the complex Klein–Gordon Lagrangian. We illustrate our methods by performing the $1/c^2$ expansion of the Kerr metric up to $\mathcal{O}(c^{-2})$, which leads to a special case of the Hartle–Thorne metric. The associated Schrödinger equation captures novel and potentially measurable effects.

# 1 Introduction

General Relativity (GR) is a well-established theory that has been thoroughly tested in many experiments [1], but all tests so far have been limited to the classical domain. Usually GR is required to describe physics at very large scales, such as in astrophysical observations or cosmology, but for tests that interface quantum mechanics, laboratory experiments are becoming increasingly relevant [2–7]. Several experimental routes have recently been proposed to test how general relativity affects the quantum dynamics and imprints signatures in genuine quantum observables at low energies

and in the weak limit beyond Newtonian gravity [8–13]. However, such tests in which GR interfaces quantum mechanics, and for which both theories are required, have not yet been realised as the relevant scales are still difficult to reach. An exception is the Newtonian limit: one class of experiments involves matter-wave superpositions in the gravitational field that experience a quantum phase shift due to the presence of the Newtonian gravitational potential [2, 4, 14–16]. Another class of such experiments are bound states in the Newtonian potential of Earth that results in a potential well and discrete energy levels for the bouncing neutrons [17, 18]. For such experiments Newtonian gravity is entirely sufficient and is typically incorporated by the addition of the Newtonian potential term in the Schrödinger equation.

With the rapid advent of ever more precise measurements of gravitational effects in quantum mechanical systems, developing a systematic framework that combines the laws of quantum mechanics with General Relativity beyond the Newtonian limit is of major interest. We stress that this is not a theory of *quantum gravity*, but rather a way to capture the gravitational effects of the background spacetime on the quantum systems. Among the myriad of applications of such a framework, let us highlight the effects of gravitational waves on quantum systems [19], post-Newtonian phase shifts [8], entanglement generated by time dilation in composite quantum systems [9], single-photon phase-shifts due to the Shapiro delay [10], decoherence of quantum superpositions due to time dilation [11], as well as quantum formulations of the Einstein equivalence principle [20]. While these effects can be described without a systematic framework, some of them have only recently been predicted due to a new-found focus on low energy systems, such as composite quantum systems in the presence of gravity beyond the Newtonian limit [9, 11, 12, 20, 21]. This has sparked renewed interest in how low-energy systems interface gravity [22–26] and how this may be probed. Such results highlight the interest in a systematic exploration of this limit, as new and overlooked effects can arise when complex quantum systems start interfacing this regime in laboratory experiments.

The purpose of this paper is to lay down the foundations of a covariant framework that utilises recent advances in nonrelativistic geometry to construct a quantum mechanical theory that takes into account gravitational effects that arise from fixed GR backgrounds. The ultimate goal is to devise a coupling prescription that gives rise to the Schrödinger equation for the centre of mass degrees of freedom of a quantum system coupled to a fixed post-Newtonian background geometry at any given order in $1/c$. For both Newtonian gravity and GR, such minimal coupling prescriptions are well-known: in the case of GR, minimal coupling instructs us to replace the Minkowski metric with the background metric $\eta \to g$, and to replace derivatives with covariant derivatives $\partial \to \nabla$. For Newtonian gravity the minimal coupling to Newton–Cartan geometry follows from coupling the wave function to the metric data and mass gauge field of Newton–Cartan geometry in a manner that respects

all the local symmetries of Newton–Cartan geometry (see, e.g., equation (3.82)). It is therefore natural to ask: what is the analogue of minimal coupling for quantum mechanics in the intermediate regime between Newton and Einstein? While a full answer to this question is likely to involve the $1/c$ expansion of the Poincaré algebra and its representations, this paper considers the[1] $1/c^2$ expansion of Lorentzian geometry and complex Klein–Gordon theory to construct a theory of quantum mechanics on post-Newtonian backgrounds order-by-order in $1/c^2$.

The time evolution of the quantum mechanical wave function $\Psi$ on $\mathbb{R}^3$ is described by the Schrödinger equation

$$\mathrm{i}\frac{\partial \Psi}{\partial t} = H\Psi \,, \tag{1.1}$$

where we set $\hbar = 1$. The Hamiltonian operator $H$ encodes the kinetic energy and the potential energy, and the simplest Hamiltonian that describes a particle of mass $m$ in a gravitational field generated by another body of mass $M$ located at the origin is

$$H = -\frac{1}{2m}\Delta - \frac{GmM}{r} \,, \tag{1.2}$$

where $G$ is Newton's constant and $\Delta$ is the Laplacian. Quantum systems described by this Hamiltonian exhibit gravitationally induced phase-shifts that have been measured with neutrons [14, 27] and atoms [2, 4, 15, 16, 28]. Such experiments confirm that the Newtonian interaction can be included in the usual quantum formalism as above, in the same way as any other potential. But one can also obtain the above Hamiltonian starting from a fully relativistic picture: Kiefer and Singh showed in [29] how the Klein-Gordon equation on curved space-time leads to the above Hamiltonian in the weak-field and nonrelativistic limit. Building on these results, Lämmerzahl studied how post-Newtonian corrections and electromagnetic interactions yield a modified Hamiltonian to first order in $c^{-2}$ in [30]. Such corrections can, for example, yield modified phase-shifts [8] which are yet to be observed. More recently, composite quantum systems have also become of interest, where the internal dynamics is affected by gravity through time-dilation and offers new prospects for experimental studies with quantum delocalised systems. The relevant coupling can be derived by simply using the mass-energy equivalence (or sometimes called the mass-defect) in the above mentioned results [9, 11, 22], as also confirmed from first-principles derivations [23, 24].

The basic lesson of General Relativity is that gravity is geometry: gravitational forces arise due to the curvature of the underlying spacetime. This remains fundamentally true for nonrelativistic gravity, where characteristic velocities are small

---

[1]For simplicity, we consider an expansion in inverse *even* powers of $c$. This is a subsector of the solution space of the full theory, which would involve a $1/c$ expansion. Not all gravitational backgrounds admit a $1/c^2$ expansion: they may contain odd powers.

compared to the speed of light. This geometric perspective is not emphasised in the approaches outlined above, but maintaining a geometric view helps highlight how fundamental GR concepts manifest themselves at the relevant scale and illuminates how unique aspects of GR affect the quantum theory. This, in turn, leads to a deeper understanding of how GR and Quantum Mechanics interface conceptually.

What does change in the nonrelativistic regime, however, is the notion of geometry. In the case of General relativity, the underlying geometry is *Lorentzian* (or pseudo-Riemannian) geometry. Nonrelativistic gravity, on the other hand, is described by non-Lorentzian geometry of Newton–Cartan type. Originally developed by Cartan more than a hundred years ago to provide a geometric framework for Newton's law of gravity [31, 32], Newton–Cartan geometry has since been generalised by considering the formal expansion of Lorentzian geometry in inverse powers of the speed of light $c$ [33–38] (see also [39, 40]). These more general geometries exist at any order in $c$ and share the same underlying *Galilean* geometric structure $(\tau_\mu, h^{\mu\nu})$ consisting of a one-form $\tau_\mu$ and a symmetric tensor $h^{\mu\nu}$ with signature $(0, 1, 1, 1)$ whose kernel is spanned by $\tau_\mu$, i.e., $h^{\mu\nu}\tau_\nu = 0$, where Greek letters represent spacetime indices, $\mu, \nu, \cdots = (t, 1, 2, 3)$. This Galilean structure is what replaces the more familiar metric $g_{\mu\nu}$ and its inverse $g^{\mu\nu}$ in Lorentzian geometry. To set up the nonrelativistic expansion, we split the metric and its inverse according to

$$g_{\mu\nu} = -c^2 T_\mu T_\nu + \Pi_{\mu\nu}\,, \qquad g^{\mu\nu} = -\frac{1}{c^2}T^\mu T^\nu + \Pi^{\mu\nu}\,, \qquad (1.3)$$

which is reminiscent of the "$3 + 1$ split" of General Relativity [41]. The components $T_\mu$, $T^\mu$, $\Pi_{\mu\nu}$ and $\Pi^{\mu\nu}$ are then expanded in inverse powers of $c$, for example,

$$\begin{aligned} T_\mu &= \tau_\mu + c^{-2}m_\mu + \mathcal{O}(c^{-4})\,, \\ \Pi^{\mu\nu} &= h^{\mu\nu} + \mathcal{O}(c^{-2})\,, \end{aligned} \qquad (1.4)$$

where we recognise the Galilean structure $(\tau_\mu, h^{\mu\nu})$ appearing at leading order (LO): the LO geometry is Galilean [37].

Here, as in the rest of this work, we expand in even inverse powers of $c$, i.e., we perform a $1/c^2$ expansion, for simplicity. As we consider higher order corrections in $1/c^2$, more and more subleading fields such as $m_\mu$ are included in the geometric description, and their transformation properties are governed by the corresponding $1/c^2$ expansion of the local Lorentz and diffeomorphisms (which can be formulated in terms of a $1/c^2$ expansion of the Poincaré algebra supplemented with appropriate curvature constraints). These higher order "gauge" fields encode gravitational effects; for example, the time component of $m_\mu$ is Newton's gravitational potential that features in (1.2).

The Schrödinger equation (1.1) is nonrelativistic in the sense discussed above: it is only valid when the particle moves slowly compared to the speed of light and the energies are lower than required for particle production. It is well known that

it is possible to turn the Klein–Gordon equation on a Lorentzian background into an equation with the same structure as the Schrödinger equation in position space $L^2(\mathbb{R}^3)$ by making a WKB-like ansatz for the Klein–Gordon field and expanding in inverse powers of $c$ [24, 26, 29, 30, 42].

Our framework builds on these works and complements them by showing that nonrelativistic geometry provides an organising principle behind these expansions, which were previously either highly specific [30] or generic [42]. It is interesting to note that the "wave functions" generated by this procedure are not wave functions in the sense of Born, since their inner product is not the standard $L^2(\mathbb{R}^3)$ norm. This is because the would-be wave functions inherit the inner product defined on (the positive-frequency part of) the Klein–Gordon solution space, and a field redefinition is required for this to reduce to the $L^2(\mathbb{R}^3)$ inner product.

For simplicity, and due to its physical relevance, we take the Galilean structure to be flat, which in Cartesian coordinates amounts to $\tau_\mu = \delta_\mu^t$ and $h^{\mu\nu} = \delta_i^\mu \delta_i^\nu$ with $i = (1,2,3)$ a spatial index. Now, both the metric and the wave function are assumed to be analytic in $1/c^2$ and hence have well-defined $1/c^2$ expansions. Including terms that are one order higher in $1/c^2$ means including three extra fields: one from the wave function and one from $T_\mu$ and $\Pi_{\mu\nu}$, respectively (cf., Eq. (1.4)). While this preponderance of fields obscures the underlying structure, their transformation properties are all inherited from the relativistic theory and follow from a $1/c^2$ expansion of the relativistic gauge symmetries. These gauge symmetries allow us to iteratively write down the Schrödinger equation coupled to a curved post-Newtonian background at any order in $1/c^2$ by making sure that all terms in the equation transform correctly under these gauge symmetries. This requires us to derive expressions for covariant derivatives at each order in $1/c^2$, which take on increasingly complicated forms as we go to higher and higher orders in $1/c^2$. This allows us, at least in principle, to write down the Schrödinger equation coupled to an arbitrary post-Newtonian background at any order in $1/c^2$.

Rather than deriving the form of the Schrödinger equation by starting from the flat space result (1.1) and requiring that it transforms correctly under gauge transformations introduced by coupling to post-Newtonian gravity order-by-order in $1/c^2$, we may also start directly from the Klein–Gordon Lagrangian and expand it in powers of $1/c^2$. At low orders in $1/c^2$, this was also considered in [37]. We show that this produces the same Schrödinger equation as our algebraic/gauge-theoretic prescription.

To illustrate our techniques in a concrete setting, we work out the nonrelativistic expansion of the Kerr metric in Boyer–Lindquist form, where in the process of the $1/c^2$ expansion we perform a coordinate transformation from oblate spherical coordinates to ordinary spherical coordinates. This leads to the Lense–Thirring metric with an additional term proportional to $J^2$ where $J$ is the angular momentum. This

metric is also the Hartle–Thorne approximation of the Kerr solution. This defines a nonrelativistic geometry to which we may couple the Schrödinger equation using the formalism that we develop. This gives rise to a quantum Hamiltonian that takes into account the gravitational effects from both the mass and the rotation. If we set the rotation equal to zero, we get the $1/c^2$ expansion of the Schwarzschild metric in Schwarzschild coordinates. These coordinates are related to isotropic coordinates via a $c$-dependent coordinate transformation, and the $1/c^2$ expansion of the Schwarzschild metric in isotropic coordinates, which are the coordinates used in [30].

The paper is structured as follows. In Section 2, we review and further develop the formalism of $1/c^2$ expansions. We show how the $1/c^2$ expansion leads to a universal Galilean structure at LO, and how the subleading fields that appear in the expansions of (1.4) encode the information of the Lorentzian spacetime to the given order in $1/c^2$. We then discuss the gauge symmetries of these fields in Section 2.2, where we also consider a flat Galilean structure.

In Section 3, we discuss how a WKB-like ansatz for the Klein–Gordon field leads to a Schrödinger-like equation, which in the limit $c \to \infty$ becomes the free Schrödinger equation. Using our results for the gauge structure of nonrelativistic geometry, we then develop a framework in Section 3.1 that allows us to derive the Schrödinger equation on a gravitational background order by order in $1/c^2$. In Section 3.4, we show show how to pass from the inner product of the Klein–Gordon fields to the $L^2(\mathbb{R}^3)$ inner product by performing a background-dependent field redefinition. We then expand the Klein–Gordon Lagrangian in Section 3.5 and demonstrate that this leads to the same equations of motion as those we obtained using bottom-up methods in Section 3.1.

We then turn our attention to an explicit example in Section 4. We begin by performing the $1/c^2$ expansion of the Kerr metric in Boyer–Lindquist coordinates in Section 4.1, leading to a generalised version of the Lense–Thirring metric which takes the form of a nonrelativistic geometry. Having identified the geometric structures, we then apply the formalism we developed in the first part of the paper to write down the Schrödinger equation on this background in Section 4.2.

We conclude with a discussion and outlook in Section 5. We have included Appendix A, which explicitly recovers previous results in the literature using the formalism we develop here. In this appendix, we furthermore discuss subtleties that arise when performing coordinate transformations that explicitly depend on $c$. We illustrate this by considering the Schrödinger equation on a Schwarzschild background expressed in either Schwarzschild or isotropic coordinates, which are related by a $c$-dependent rescaling of the radial direction.

## 2  Nonrelativistic expansion of spacetime geometry

It is well known that the nonrelativistic limit of a relativistic theory is obtained by expanding in inverse powers of the speed of light $c$. Rather than coupling to familiar Lorentzian spacetimes, i.e., pseudo-Riemannian geometries of signature $(-1, 1, 1, 1)$ in four spacetime dimensions, these expanded theories couple to spacetimes that arise by expanding the Lorentzian spacetimes in $1/c$.

In this section, we expand Lorentzian geometry in powers of $1/c^2$. Such systematic expansions in inverse powers of the speed of light were considered in [33] (see also [39, 40]), and, more recently, an appropriately truncated expansion of the expanded geometry was used to write down an action for nonrelativistic gravity [34, 36, 37] (see also the review [38]). These geometries do not possess a Lorentzian metric, but rather come equipped with a *Galilean* structure consisting of a nowhere vanishing corank one "spatial metric" and a nowhere vanishing "clock" one-form. These geometries generalise Newton–Cartan geometry, which was originally conceived by Cartan [31, 32] (see e.g. [43, Ch. 12] for a pedagogical introduction) to provide a geometric framework in which to formulate Newton's equations of motion in a covariant way, just as Lorentzian geometry provides the geometric framework underlying Einstein's equation. To distinguish this original Newton–Cartan geometry from the one employed in the formulation of off-shell nonrelativistic gravity [34, 36, 37], the latter geometry was dubbed "type II torsional Newton–Cartan geometry".

### 2.1  $1/c^2$ expansion of Lorentzian geometry

Consider a $(d+1)$-dimensional[2] manifold $M$ equipped with a Lorentzian metric $g_{\mu\nu}$ $(\mu, \nu = 0, 1, \ldots, d)$. We split the metric into timelike and spacelike components as follows

$$g_{\mu\nu} = -c^2 T_\mu T_\nu + \Pi_{\mu\nu} \,, \tag{2.1}$$

with a similar relation holding for the inverse metric $g^{\mu\nu}$

$$g^{\mu\nu} = -\frac{1}{c^2} T^\mu T^\nu + \Pi^{\mu\nu} \,. \tag{2.2}$$

The objects $T_\mu$ and $\Pi_{\mu\nu}$ and their inverses satisfy the relations

$$T_\mu \Pi^{\mu\nu} = 0 = T^\mu \Pi^{\mu\nu} \,, \qquad T_\mu T^\mu = -1 \,, \qquad \delta^\nu_\mu = -T^\nu T_\mu + \Pi_{\mu\rho}\Pi^{\rho\nu} \,. \tag{2.3}$$

We emphasise that this still describes a Lorentzian structure: the above is just a reparameterisation of the metric $g_{\mu\nu}$ and its inverse. To turn this into a "nonrelativistic" (NR) geometry, we formally Taylor expand the fields $T$ and $\Pi$ in powers

---

[2]While the analysis in this section is performed in general spacetime dimension, we will later specialise to the physically relevant four-dimensional spacetimes.

of $1/c^2$.[3] Note that concrete applications of this scheme requires the existence of a suitable characteristic velocity $v_{\rm ch} \ll c$ such that the formal $1/c^2$ expansion turns into an expansion in the dimensionless parameter $\epsilon = v_{\rm ch}^2/c^2$. Hence, the geometric fields $T_\mu$ and $\Pi_{\mu\nu}$ are expanded as

$$
\begin{aligned}
T_\mu &= \tau_\mu + c^{-2}m_\mu + c^{-4}B_\mu + \mathcal{O}(c^{-6})\,, \\
\Pi_{\mu\nu} &= h_{\mu\nu} + c^{-2}\Phi_{\mu\nu} + \mathcal{O}(c^{-4})\,.
\end{aligned}
\tag{2.4}
$$

Here at each order new fields are introduced, which will be discussed further below. The field $\tau_\mu$ is known as the clock 1-form and measures the proper time $\mathcal{T}$ along any curve $\gamma$ in the resulting nonrelativistic geometry

$$
\mathcal{T} = \int_\gamma \tau_\mu dx^\mu\,.
\tag{2.5}
$$

When $\tau \wedge d\tau = 0$, in which case $\tau$ gives rise to a foliation in terms of hypersurfaces of absolute simultaneity, the field $h_{\mu\nu}$ measures spatial distances on these hypersurfaces when pulled back to the leaves of the foliation. The condition $\tau \wedge d\tau = 0$ is implied by the Einstein equations. The expansions of $T_\mu$ and $\Pi_{\mu\nu}$ mean that the metric expands according to [37]

$$
g_{\mu\nu} = -c^2\tau_\mu\tau_\nu + \bar{h}_{\mu\nu} + c^{-2}\bar{\Phi}_{\mu\nu} + \mathcal{O}(c^{-4})\,,
\tag{2.6}
$$

where

$$
\bar{h}_{\mu\nu} = h_{\mu\nu} - 2\tau_{(\mu}m_{\nu)}\,,
\tag{2.7a}
$$

$$
\bar{\Phi}_{\mu\nu} = \Phi_{\mu\nu} - m_\mu m_\nu - 2B_{(\mu}\tau_{\nu)}\,.
\tag{2.7b}
$$

For the inverse structures, we have similar expansions

$$
\begin{aligned}
T^\mu &= v^\mu + c^{-2}X^\mu + c^{-4}Y^\mu + \mathcal{O}(c^{-6})\,, \\
\Pi^{\mu\nu} &= h^{\mu\nu} + c^{-2}P^{\mu\nu} + c^{-4}Q^{\mu\nu} + c^{-6}W^{\mu\nu} + \mathcal{O}(c^{-8})\,.
\end{aligned}
\tag{2.8}
$$

The relations (2.3) imply that the leading order (LO) fields satisfy

$$
v^\mu\tau_\mu = -1\,, \qquad v^\mu h_{\mu\nu} = \tau_\mu h^{\mu\nu} = 0\,, \qquad \delta^\mu_\nu = -v^\mu\tau_\nu + h^{\mu\rho}h_{\rho\nu}\,.
\tag{2.9}
$$

Together, the fields $(\tau_\mu, h^{\mu\nu})$ define a *Galilean* structure. As we will see in Section 2.2, these fields are inert under local tangent space transformations. Subleading fields, such as $m_\mu$ and $\Phi_{\mu\nu}$ can be considered as "gauge fields" that are defined on a nonrelativistic spacetime. These subleading fields are part of the $1/c^2$ corrected geometry and are dynamical fields in a theory of nonrelativistic gravity [37]. A Galilean structure is entirely determined by the properties of the clock form [45–48]; we will be

---

[3]In principle, we should also include odd powers, but we leave them out for simplicity. See [44] for a treatment of such terms in the context of gravity.

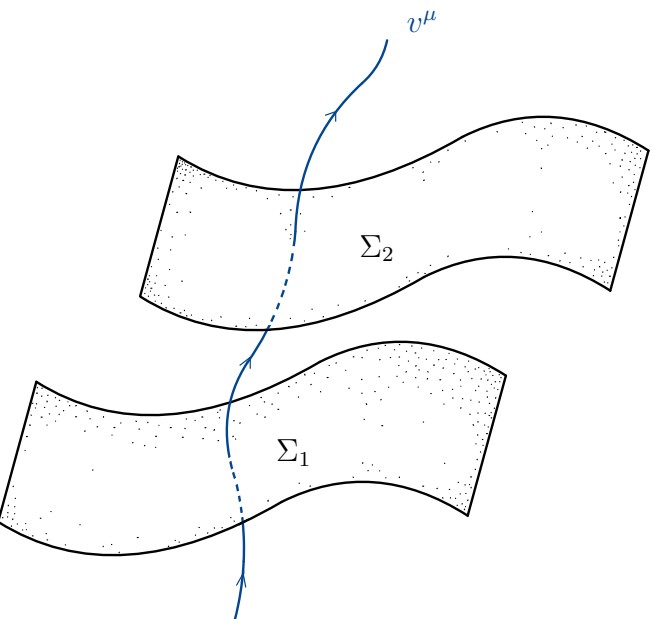

**Figure 1**. A cartoon of Newton–Cartan geometry. When $d\tau = 0$, there exists an absolute time $t$. Then $\Sigma_1$ and $\Sigma_2$ are equal-time hypersurfaces which are equipped with a Riemannian metric, namely $h_{\mu\nu}$ restricted to the spatial surface. The Newton–Cartan "velocity" $v^\mu$ is an observer-dependent (since it transforms under Galilean boosts) vector field that is orthogonal to the equal-time hypersurfaces.

interested in the case when $\tau$ is (locally) exact in which case there exists a notion of absolute time: that is, the proper time $\mathcal{T}$ in (2.5) between any two points in the nonrelativistic spacetime is the same regardless of the curve $\gamma$ that connects them (see Figure 1). An exact clock form is required to obtain the Newtonian limit of GR [37]. In fact, we will see in Section 3.5 that, at least in the absence of an external electromagnetic field, the clock form is determined by the WKB phase that defines the relation between the Klein–Gordon field and the nonrelativistic wave function. This observation was also made in [37], where it was shown that various (bosonic) matter field theories, including electromagnetism, have actions that expand in such a way that no torsion is generated.

The relations (2.3) furthermore imply that the subleading fields that appear in $T^\mu$ and $\Pi^{\mu\nu}$ are entirely determined by the subleading fields that appear in $T_\mu$ and

$\Pi_{\mu\nu}$. Explicitly, we have

$$
\begin{aligned}
X^\mu &= -v^\mu \Phi - h^{\mu\rho} v^\nu \Phi_{\nu\rho}\,, \\
Y^\mu &= v^\mu \Phi^2 - v^\mu m_\nu h^{\nu\rho} v^\sigma \Phi_{\sigma\rho} + v^\mu v^\nu B_\nu + h^{\mu\rho} X_\rho\,. \\
P^{\mu\nu} &= 2v^{(\mu} h^{\nu)\rho} m_\rho - h^{\mu\rho} h^{\nu\sigma} \Phi_{\rho\sigma}\,, \\
Q^{\mu\nu} &= v^\mu v^\nu h^{\rho\sigma} m_\rho m_\sigma + 2v^{(\mu} h^{\nu)\rho} B_\rho - 2\Phi v^{(\mu} h^{\nu)\rho} m_\rho \\
&\quad - 2v^{(\mu} h^{\nu)\sigma} h^{\rho\lambda} m_\rho \Phi_{\lambda\sigma} + h^{\mu\rho} h^{\nu\sigma} X_{\rho\sigma}\,, \\
W^{\mu\nu} &= v^\mu v^\nu \left[ 2h^{\rho\sigma} B_\rho m_\sigma - 2\Phi h^{\rho\sigma} m_\rho m_\sigma - h^{\rho\lambda} h^{\sigma\kappa} m_\rho m_\sigma \Phi_{\lambda\kappa} \right] \\
&\quad + v^\mu h^{\nu\rho} \tilde{X}_\rho + v^\nu h^{\mu\rho} \tilde{X}_\rho + h^{\mu\rho} h^{\nu\sigma} \tilde{X}_{\rho\sigma}\,,
\end{aligned}
\tag{2.10}
$$

where we will not need the explicit forms of $X_\rho$, $\tilde{X}_\mu$, $X_{\rho\sigma}$, $\tilde{X}_{\rho\sigma}$ and where we defined

$$
\Phi = -v^\mu m_\mu\,.
\tag{2.11}
$$

In deriving these expressions, we have used the fact that the LO relations (2.9) imply that any contravariant 2-tensor $X^{\mu\nu}$ may be decomposed as

$$
X^{\mu\nu} = \tau_\rho \tau_\lambda v^\mu v^\nu X^{\rho\lambda} + h^{\mu\sigma} h_{\sigma\rho} h^{\nu\kappa} h_{\kappa\lambda} X^{\rho\lambda} - 2v^{(\mu} h^{\nu)\sigma} h_{\sigma\rho} \tau_\lambda X^{\rho\lambda}\,.
\tag{2.12}
$$

In Section 3.5, we will consider the $1/c^2$ expansion of actions defined on Lorentzian backgrounds, which involve the integration measure $\sqrt{-g} d^{d+1}x$, where $g = \det(g_{\mu\nu})$ is the determinant of the metric. For the square root of the determinant of the metric, we write

$$
\sqrt{-\det(g)} = cE\,,
\tag{2.13}
$$

where $E$ expands in powers of $1/c^2$ as

$$
E = e\left(1 + c^{-2}\left[\Phi + \frac{1}{2} h^{\mu\nu}\Phi_{\mu\nu}\right]\right) + \mathcal{O}(c^{-4})\,,
\tag{2.14}
$$

where $e = \sqrt{\det(\tau_\mu \tau_\nu + h_{\mu\nu})}$ defines the integration measure $e d^{d+1}x$ of the Galilean structure.

## 2.2 Gauge structure & flat LO geometry

The subleading fields that appear in the expansion of $T_\mu$ and $\Pi_{\mu\nu}$ up to (and including) next-to-leading order (NLO) are $\tau_\mu, h_{\mu\nu}, m_\mu, \Phi_{\mu\nu}$. These data define a type II torsional Newton–Cartan geometry [34, 37]. In order to determine the metric at order $c^{-2}$ we need to include the NNLO field $B_\mu$ from the expansion of $T_\mu$. In this section, we work out the transformation properties of these fields, which will allow us to uniquely fix the structure order-by-order of the Schrödinger equation coupled to a nonrelativistic geometry up to a given order in $1/c^2$. In particular, the local tangent space symmetries of $(d+1)$-dimensional Lorentzian geometry form the Lorentz

group $SO(d,1)$. By expanding the corresponding Lie algebra $\mathfrak{so}(d,1)$ in powers of $1/c^2$, one obtains, after suitable quotienting, the local tangent space algebra of the nonrelativistic geometry at that order (see [34, 37, 49] for more details).

To elucidate the local tangent space structure, it is useful to decompose $\Pi_{\mu\nu}$ that appears in the decomposition (2.1) in terms of spatial vielbeine $\mathcal{E}_\mu^a$ as

$$\Pi_{\mu\nu} = \delta_{ab}\mathcal{E}_\mu^a\mathcal{E}_\nu^b\,, \tag{2.15}$$

where $a,b = 1,\ldots,d$ are spatial tangent space indices. The vielbeine have a $1/c^2$ expansion of the form

$$\mathcal{E}_\mu^a = e_\mu^a + c^{-2}\pi_\mu^a + \mathcal{O}(c^{-4})\,, \tag{2.16}$$

which means that

$$h_{\mu\nu} = \delta_{ab}e_\mu^a e_\nu^b\,, \qquad \Phi_{\mu\nu} = 2\delta_{ab}e_{(\mu}^a\pi_{\nu)}^b\,, \tag{2.17}$$

which implies

$$v^\mu v^\nu\Phi_{\mu\nu} = 0\,. \tag{2.18}$$

The metric transforms under diffeomorphisms infinitesimally generated by a vector $\Xi^\mu$ as $\delta_\Xi g_{\mu\nu} = \mathcal{L}_\Xi g_{\mu\nu}$, where $\mathcal{L}$ denotes the Lie derivative. The vector $\Xi^\mu$ has a $1/c^2$ expansion of the form [37]

$$\Xi^\mu = \xi^\mu + c^{-2}\zeta^\mu + c^{-4}\kappa^\mu + \mathcal{O}(c^{-6})\,. \tag{2.19}$$

We are demanding that the diffeomorphisms preserve the $1/c^2$ expansion properties of the metric.

The LO diffeomorphism $\xi^\mu$ will behave as diffeomorphisms in the nonrelativistic geometry, while the subleading diffeomorphisms will not: instead, they admit an interpretation as extra gauge symmetries in the theory. In addition to diffeomorphisms, the vielbeine $T_\mu$ and $\mathcal{E}_\mu^a$ transform under local Lorentz transformations $(\Lambda^a{}_b, \Lambda^a)$, where $\Lambda^a{}_b$ is a local rotation and $\Lambda^a$ is a local boost, as [37]

$$\begin{aligned}
\delta_\Lambda T_\mu &= c^{-2}\Lambda_a\mathcal{E}_\mu^a\,, \\
\delta_\Lambda\mathcal{E}_\mu^a &= \Lambda^a T_\mu + \Lambda^a{}_b\mathcal{E}_\mu^b\,.
\end{aligned} \tag{2.20}$$

The local rotations and boosts have $1/c^2$ expansions of the form

$$\begin{aligned}
\Lambda^a &= \lambda^a + c^{-2}\eta^a + \mathcal{O}(c^{-4})\,, \\
\Lambda^a{}_b &= \lambda^a{}_b + c^{-2}\sigma^a{}_b + \mathcal{O}(c^{-4})\,,
\end{aligned} \tag{2.21}$$

where $\lambda^a{}_b$ is a local rotation in the nonrelativistic geometry, while $\lambda^a$ is a Galilean boost. Again, we are assuming that the local Lorentz transformations preserve the

$1/c^2$ expansion of the vielbeine. The subleading boosts $\eta^a$ and rotations $\sigma^a{}_b$ act as gauge symmetries on the NLO fields. Combining all these transformations, we get

$$
\begin{aligned}
\delta\tau_\mu &= \mathcal{L}_\xi \tau_\mu \,, \\
\delta h_{\mu\nu} &= \mathcal{L}_\xi h_{\mu\nu} + 2\lambda_a e^a_{(\mu}\tau_{\nu)} \,, \\
\delta m_\mu &= \mathcal{L}_\xi m_\mu + \mathcal{L}_\zeta \tau_\mu + \lambda_a e^a_\mu \,, \\
\delta\Phi_{\mu\nu} &= \mathcal{L}_\xi \Phi_{\mu\nu} + \mathcal{L}_\zeta h_{\mu\nu} + 2\lambda_a e^a_{(\mu}m_{\nu)} + 2\lambda_a \pi^a_{(\mu}\tau_{\nu)} + 2\eta_a e^a_{(\mu}\tau_{\nu)} \,, \\
\delta B_\mu &= \mathcal{L}_\xi B_\mu + \mathcal{L}_\zeta m_\mu + \mathcal{L}_\kappa \tau_\mu + \eta_a e^a_\mu + \lambda_a \pi^a_\mu \,.
\end{aligned}
\tag{2.22}
$$

At this stage, it is useful to introduce an inverse vielbein $e^\mu_a$ to $e^a_\mu$, which satisfies $e^\mu_a \tau_\mu = 0$ and $e^a_\mu e^\mu_b = \delta^a_b$. We then have $h^{\mu\nu} = \delta^{ab} e^\mu_a e^\nu_b$. In terms of the vielbeine $e^a_\mu$ and $e^\mu_a$, the completeness relation reads $e^\nu_a e^a_\mu - v^\nu \tau_\mu = \delta^\nu_\mu$, which we can use to write $\lambda_a \pi^a_\mu = \lambda_a \pi^a_\nu e^\nu_b e^b_\mu - \lambda_a \pi^a_\nu v^\nu \tau_\mu$. Hence, we find that

$$
\eta_a e^a_\mu + \lambda_a \pi^a_\mu = \eta_a e^a_\mu + \lambda_a \pi^a_\nu e^\nu_b e^b_\mu - \lambda_a \pi^a_\nu v^\nu \tau_\mu = \tilde{\eta}_a e^a_\mu - \lambda_\rho h^{\rho\sigma} v^\kappa \Phi_{\sigma\kappa}\tau_\mu \,,
\tag{2.23}
$$

where $\tilde{\eta}_a = \eta_a + \lambda_b \pi^b_\nu e^\nu_a$ and where we used

$$
\lambda_\rho h^{\rho\sigma} v^\kappa \Phi_{\sigma\kappa} = \lambda_a \pi^a_\rho v^\rho \,,
\tag{2.24}
$$

which follows from (2.17) and the definition $\lambda_\mu = \lambda_a e^a_\mu$. We can use this to eliminate $\pi^a_\mu$ from the gauge transformations of $\Phi_{\mu\nu}$ and $B_\mu$ in favour of $\Phi_{\mu\nu}$ since it allows us to write

$$
\begin{aligned}
\delta\Phi_{\mu\nu} &= \mathcal{L}_\xi \Phi_{\mu\nu} + \mathcal{L}_\zeta h_{\mu\nu} + 2\lambda_a e^a_{(\mu}m_{\nu)} + 2\tilde{\eta}_a e^a_{(\mu}\tau_{\nu)} - 2\lambda_\rho h^{\rho\sigma} v^\kappa \Phi_{\sigma\kappa}\tau_\mu \tau_\nu \,, \\
\delta B_\mu &= \mathcal{L}_\xi B_\mu + \mathcal{L}_\zeta m_\mu + \mathcal{L}_\kappa \tau_\mu + \tilde{\eta}_a e^a_\mu - \lambda_\rho h^{\rho\sigma} v^\kappa \Phi_{\sigma\kappa}\tau_\mu \,.
\end{aligned}
\tag{2.25}
$$

We will assume throughout that the clock one-form $\tau$ is closed, i.e.

$$
(d\tau)_{\mu\nu} = \partial_\mu \tau_\nu - \partial_\nu \tau_\mu = 0 \,.
\tag{2.26}
$$

Since we will not allow for non-contractible closed timelike loops, this is equivalent to saying that $\tau$ is exact, i.e., that time is absolute. Using that $d\tau = 0$ the transformations in (2.22) can be written as

$$
\delta\tau_\mu = \partial_\mu \left(\tau_\nu \xi^\nu\right) \,,
\tag{2.27a}
$$

$$
\delta h_{\mu\nu} = \mathcal{L}_\xi h_{\mu\nu} + 2\lambda_{(\mu}\tau_{\nu)} \,,
\tag{2.27b}
$$

$$
\delta m_\mu = \mathcal{L}_\xi m_\mu + \partial_\mu \Lambda + \lambda_\mu \,,
\tag{2.27c}
$$

$$
\delta\Phi_{\mu\nu} = \mathcal{L}_\xi \Phi_{\mu\nu} + \mathcal{L}_\zeta h_{\mu\nu} + 2\lambda_{(\mu}m_{\nu)} - 2\lambda_\rho h^{\rho\sigma} v^\kappa \Phi_{\sigma\kappa}\tau_\mu \tau_\nu + 2\tilde{\eta}_{(\mu}\tau_{\nu)} \,,
\tag{2.27d}
$$

$$
\delta B_\mu = \mathcal{L}_\xi B_\mu + \mathcal{L}_\zeta m_\mu + \partial_\mu \chi + \tilde{\eta}_\mu - \lambda_\rho h^{\rho\sigma} v^\kappa \Phi_{\sigma\kappa}\tau_\mu \,,
\tag{2.27e}
$$

where we defined

$$
\begin{aligned}
\Lambda &= \tau_\nu \zeta^\nu \,, \\
\chi &= \tau_\nu \kappa^\nu \,, \\
\tilde{\eta}_\mu &= \tilde{\eta}_a e^a_\mu \,.
\end{aligned}
\tag{2.28}
$$

This is the form of the gauge transformations that we will work with in what follows.

We will furthermore assume that the LO geometry described by $\tau_\mu$ and $h^{\mu\nu}$ is a flat. This means that we can go to a Cartesian coordinate system in which

$$h^{\mu\nu} = \delta_i^\mu \delta_j^\nu \delta^{ij} \,, \qquad h_{\mu\nu} = \delta_\mu^i \delta_\nu^j \delta_{ij} \,, \qquad \tau_\mu = \delta_\mu^t \,, \qquad v^\mu = -\delta_t^\mu \,, \qquad (2.29)$$

where we split the spacetime index according to $\mu = (t, i)$, where now $i, j, k, \cdots = 1, \ldots, d$ are spatial indices.

The residual gauge transformations of this gauge choice are all the transformations for which $\delta\tau_\mu = 0$ and $\delta h_{\mu\nu} = 0$ where the transformation is given in (2.27). This means that $\xi^t = \text{cst}$, $\lambda_t = 0$, $\lambda_i = -\partial_t \xi^i$ and $\partial_i \xi^j + \partial_j \xi^i = 0$. The latter equation can be solved by hitting it with $\partial_k$, leading to $\partial_k (\partial_i \xi_j + \partial_j \xi_i) = 0$, where $\xi^i = \xi_i$. For the flat spatial geometry, where indices are raised and lowered by a Kronecker delta, we do not distinguish between raised and lowered indices. Next, we write down all three cyclic permutations of this equation by permuting the indices $i, j, k$. Adding two of these and subtracting the third leads to $\partial_k (\partial_j \xi_i - \partial_i \xi_j) = 0$. Adding this to $\partial_k (\partial_i \xi_j + \partial_j \xi_i) = 0$ leads to $\partial_k \partial_j \xi_i = 0$. This equation can be solved to give

$$\xi^i = a^i(t) + \lambda^i{}_j(t) x^j \,. \qquad (2.30)$$

This solves $\partial_i \xi^j + \partial_j \xi^i = 0$ provided $\lambda^i{}_j = -\lambda^j{}_i$. The residual gauge transformations are thus time-dependent translations $a^i(t)$ and time-dependent rotations $\lambda^i{}_j(t)$. The vectors $\xi = \xi^t \partial_t + \xi^i \partial_i$ with $(\xi^t, \xi^i)$ as above are Killing vectors in the sense that they obey $\mathcal{L}_\xi \tau_\mu = 0 = \mathcal{L}_\xi h^{\mu\nu} = 0$. These Killing vectors form the Coriolis algebra [50].

Omitting the time-dependent rotations, it follows from (2.27) that the subleading fields for a flat LO geometry transform as

$$\delta m_t = a^i \partial_i m_t + m_i \partial_t a^i + \partial_t \Lambda \,, \qquad (2.31a)$$

$$\delta m_i = a^j \partial_j m_i - \partial_t a^i + \partial_i \Lambda \,, \qquad (2.31b)$$

$$\delta \Phi_{it} = a^j \partial_j \Phi_{it} + \Phi_{ij} \partial_t a^j + \tilde{\eta}_i - m_t \partial_t a^i + \partial_t \zeta_i \,, \qquad (2.31c)$$

$$\delta \Phi_{ij} = a^k \partial_k \Phi_{ij} - m_i \partial_t a^j - m_j \partial_t a^i + 2\partial_{(i} \zeta_{j)} \,, \qquad (2.31d)$$

$$\delta B_t = a^i \partial_i B_t + B_i \partial_t a^i - \Phi_{it} \partial_t a^i + \Lambda \partial_t m_t + m_t \partial_t \Lambda + \zeta^i \partial_i m_t + m_i \partial_t \zeta^i + \partial_t \chi \,, \qquad (2.31e)$$

$$\delta B_i = a^j \partial_j B_i + \tilde{\eta}_i + \Lambda \partial_t m_i + m_t \partial_i \Lambda + \zeta^j \partial_j m_i + m_j \partial_i \zeta^j \,, \qquad (2.31f)$$

where $a^i$ only depends on $t$ and where $\Lambda$, $\tilde{\eta}_i$, $\zeta_i$ and $\chi$ are arbitrary. We note that we can always set $\Phi_{it} = 0$ by fixing the $\tilde{\eta}_i$ gauge transformation which describes a subleading local boost. The residual gauge transformations have an $\tilde{\eta}_i$ that can be solved by setting $\delta\Phi_{it} = 0$. This however makes the transformation of $B_i$ a bit more complicated, and hence we will refrain from doing this.

These transformations will play a key rôle in the next section. The next ingredient we need is the notion of a complex scalar field, i.e., the wave function, that is

defined on the flat spacetime as described above. This requires that we understand the $1/c^2$ corrections to the wave function as well as the transformation properties under diffeomorphisms and subleading diffeomorphisms. This allows us to define covariant derivatives acting on the wave function from which we can build equations of motion that couple the wave function to the $1/c^2$ expanded geometry. This will be the subject of the next section.

# 3  Gravitational corrections in quantum mechanics

It is well known that a complex scalar $\phi_{\text{KG}}$ that obeys the Klein–Gordon equation in Minkowski space gives rise to a wave function $\Psi$ that satisfies the Schrödinger equation upon making the decomposition [24, 26, 29, 30, 42, 51]

$$\phi_{\text{KG}} = e^{-imc^2 t}\Psi\,, \tag{3.1}$$

where $m$ is the mass of the complex scalar, which also becomes the mass of the Schrödinger field in the nonrelativistic quantum mechanics picture. The Galilean absolute time $t$ that appears in the exponential factor defines the clock form via

$$\tau = dt\,. \tag{3.2}$$

In Minkowski space, the Klein–Gordon equation for a free scalar field reads

$$\eta^{\mu\nu}\partial_\mu\partial_\nu\phi_{\text{KG}} - m^2 c^2\phi_{\text{KG}} = 0\,, \tag{3.3}$$

where $\eta^{\mu\nu} = (-c^{-2}, \delta^{ij})$ is the (inverse) Minkowski metric. Using the decomposition (3.1), the equation above becomes

$$i\partial_t\Psi = -\frac{1}{2m}\partial_i\partial_i\Psi + \frac{1}{2mc^2}\partial_t^2\Psi\,. \tag{3.4}$$

If we expand the field $\Psi = \psi_{(0)} + c^{-2}\psi_{(2)} + \mathcal{O}(c^{-4})$ we obtain the LO and NLO Schrödinger equations

$$i\partial_t\psi_{(0)} = -\frac{1}{2m}\partial_i\partial_i\psi_{(0)}\,, \tag{3.5}$$

$$i\partial_t\psi_{(2)} = -\frac{1}{2m}\partial_i\partial_i\psi_{(2)} + \frac{1}{2m}\partial_t^2\psi_{(0)}\,. \tag{3.6}$$

In the rest of this section we will design a coupling prescription that allows us to couple these equations to a NC geometry plus its $1/c^2$ correction.

## 3.1  LO Schrödinger equation

In order to describe modifications to Schrödinger's equation (3.5) due to relativistic effects and gravity, we must include $1/c^2$ corrections in its formulation. In this

section, we develop a framework that allows us to obtain Schrödinger's equation using the geometric framework developed in Section 2. We make the simplifying assumption that the Galilean structure is flat, cf., (2.29). As we discussed above, the inclusion of $1/c^2$ corrections implies that $\Psi$ admits an expansion of the form

$$\Psi = \psi_{(0)} + c^{-2}\psi_{(2)} + \mathcal{O}(c^{-4}) \,. \tag{3.7}$$

The Klein–Gordon field $\phi_{\mathrm{KG}}$ transforms under infinitesimal diffeomorphisms generated by $\Xi^\mu$ (cf., (2.19)) as

$$\delta\phi_{\mathrm{KG}} = \Xi^\mu \partial_\mu \phi_{\mathrm{KG}} \,, \tag{3.8}$$

where $\Xi^\mu$ expands as in (2.19). As we saw in Section 2.2, the residual temporal LO diffeomorphisms $\xi^t$ that preserve the LO Cartesian structure (2.29) (see also (3.2) above) are just constant shifts. In what follows, we take $\xi^t = 0$ since this particular transformation will not be useful when fixing the form of the Schrödinger equation. This means that the decomposition of the Klein–Gordon field in (3.1), and its expansion in (3.7), combined with the transformation (3.8) and the expansion (2.19), lead to

$$\begin{aligned}
\delta\psi_{(0)} &= a^i \partial_i \psi_{(0)} - \mathrm{i}m\Lambda\psi_{(0)} \,, \\
\delta\psi_{(2)} &= a^i \partial_i \psi_{(2)} - \mathrm{i}m\Lambda\psi_{(2)} + \Lambda\partial_t\psi_{(0)} - \mathrm{i}m\chi\psi_{(0)} + \zeta^i\partial_i\psi_{(0)} \,,
\end{aligned} \tag{3.9}$$

where $\Lambda$ and $\chi$ are defined in (2.28) and where we used (2.30). We also used that the global time $t$ is inert (since we took $\xi^t = 0$), and where we omitted the time-dependent rotations $\lambda^i{}_j(t)$ since they will not be needed in what follows. The transformations of $\psi_{(0)}$ and $\psi_{(2)}$ are of course such that

$$\phi_{\mathrm{KG}} = \mathrm{e}^{-\mathrm{i}mc^2 t}\left(\psi_{(0)} + c^{-2}\psi_{(2)} + \mathcal{O}(c^{-4})\right) \tag{3.10}$$

transforms like a scalar field under general $c$-dependent coordinate transformations.

These transformations can also be understood as follows. Under a global time translation $t' = t + t_0$, $x'^i = x^i$ we have

$$\Psi'(t,x) = \mathrm{e}^{\mathrm{i}mc^2 t_0}\Psi(t - t_0, x) \,. \tag{3.11}$$

If we assume that $t_0$ is small, then to first order in $t_0$ we obtain

$$\delta\Psi(t,x) = \Psi'(t,x) - \Psi(t,x) = \mathrm{i}mc^2 t_0 \Psi(t,x) - t_0 \Psi'(t,x) \,. \tag{3.12}$$

If we gauge this symmetry by replacing $t_0$ with $-\Xi^t$ and expand the latter in $1/c^2$ as follows

$$\Xi^t = c^{-2}\Lambda + c^{-4}\chi + \mathcal{O}(c^{-6}) \,, \tag{3.13}$$

we obtain (3.9) (with $a^i = 0 = \zeta^i$) where we also expanded $\Psi$ in $c^{-2}$. In the expansion of $\Xi^t$ we omitted the LO term $\xi^t$ since this was also not considered in (3.9) as this is just a constant since we are not changing coordinates at LO. What this shows is that the appearance of the $\Lambda$ and $\chi$ terms in (3.9) is due to time reparametrisations in GR. Similarly, the presence of $\zeta^i$ (and $a^i$) is dictated by spatial coordinate transformations in GR. The field $\psi_{(0)}$ transforms as a complex scalar field with respect to the LO diffeomorphisms and it has a (linear) local $U(1)$ transformation acting on it whose parameter $\Lambda$ comes from NLO time reparametrisations (see [52] for related observations). The $U(1)$-like transformations with parameters $\{\Lambda, \chi\}$ will play an important rôle in the construction of suitable gauge covariant derivatives, which allow for a natural formulation of Schrödinger's equation coupled to the $1/c^2$ expanded geometries described in Section 2.

The Schrödinger equation for $\psi_{(0)}$ can be formulated as $O\psi_{(0)} = 0$ where $O$ is an operator that does not depend on $\psi_{(0)}$. The object $O\psi_{(0)}$ (without setting it to zero) should transform like $\psi_{(0)}$. We will denote $O\psi_{(0)}$ by "LO Eq.", i.e., the leading order equation, and since this should transform like $\psi_{(0)}$ we demand that

$$\delta(\text{LO Eq.}) = a^i \partial_i (\text{LO Eq.}) - im\Lambda(\text{LO Eq.}) \,. \tag{3.14}$$

The LO Schrödinger equation should contain (3.5). In other words, we must construct an object $X$ that enters the LO equation as

$$\text{LO Eq.} = i\partial_t \psi_{(0)} + \frac{1}{2m}\partial_i \partial_i \psi_{(0)} + X \,, \tag{3.15}$$

in such a way that (3.14) holds.

In order to construct this $X$, it is useful to introduce a gauge covariant derivative $\mathcal{D}_\mu$ that acts on the LO wave function $\psi_{(0)}$ as

$$\mathcal{D}_\mu \psi_{(0)} := \partial_\mu \psi_{(0)} + imm_\mu \psi_{(0)} \,. \tag{3.16}$$

The combination $\mathcal{D}_\mu \psi_{(0)}$ transforms as

$$\delta_\Lambda \mathcal{D}_\mu \psi_{(0)} = -im\Lambda \mathcal{D}_\mu \psi_{(0)} \,, \tag{3.17a}$$

$$\delta_a \mathcal{D}_t \psi_{(0)} = a^i \partial_i (\mathcal{D}_t \psi_{(0)}) + (\partial_t a^i)\mathcal{D}_i \psi_{(0)} \,, \tag{3.17b}$$

$$\delta_a \mathcal{D}_i \psi_{(0)} = a^j \partial_j (\mathcal{D}_i \psi_{(0)}) - im\psi_{(0)}\partial_t a^i \,, \tag{3.17c}$$

where we used equations (2.31a), (2.31b) and (3.9). By the $\delta_a$ transformation we mean all the $a^i$-dependent terms in the transformations of (2.31a), (2.31b) and (3.9). We note that the $a^i$-dependent terms have two origins: one is from Lie derivatives with respect to residual LO diffeomorphisms acting on the gauge field $m_\mu$ and the other is from the residual Galilean boosts with parameter $\lambda_t = 0$, $\lambda_i = -\partial_t a^i$. When we say the derivative $\mathcal{D}_\mu$ is covariant we mean here with respect to the $\Lambda$ gauge transformation.

We also need the double spatial covariant derivative

$$\mathcal{D}_i \mathcal{D}_j \psi_{(0)} = \partial_i \mathcal{D}_j \psi_{(0)} + \mathrm{i} m m_i \mathcal{D}_j \psi_{(0)} \,, \tag{3.18}$$

which transforms as

$$\delta_\Lambda \mathcal{D}_i \mathcal{D}_j \psi_{(0)} = -\mathrm{i} m \Lambda \mathcal{D}_i \mathcal{D}_j \psi_{(0)} \,, \tag{3.19a}$$
$$\delta_a \mathcal{D}_i \mathcal{D}_j \psi_{(0)} = a^k \partial_k (\mathcal{D}_i \mathcal{D}_j \psi_{(0)}) - 2\mathrm{i} m \partial_t a_{(j} \mathcal{D}_{i)} \psi_{(0)} \,. \tag{3.19b}$$

The usefulness of the covariant derivative (3.16) stems from the property that it is constructed precisely such that if we replace all ordinary derivatives in (3.5) with covariant derivatives, we automatically make sure that the LO equation transforms covariantly under $\Lambda$ transformations. Thus, the equation

$$\text{LO Eq.} = \mathrm{i} \mathcal{D}_t \psi_{(0)} + \frac{k}{2m} \mathcal{D}_i \mathcal{D}_i \psi_{(0)} \,, \tag{3.20}$$

where $k$ is a real constant that will be fixed shortly, transforms correctly under $\Lambda$-transformations. We must also check that the LO equation transforms correctly under time-dependent translations $a^i$ (that preserve the frame choice $h_{it} = 0$ which is affected by a compensating local Galilean boost transformation with $\lambda_i = -\partial_t a^i$), and using (3.17b) and (3.19b), we find that the LO equation (3.20) transforms as

$$\delta_a(\text{LO Eq.}) = a^j \partial_j(\text{LO Eq.}) \,, \tag{3.21}$$

provided we take $k = 1$, in accordance with (3.14). This means that $X$ in (3.15) is given by

$$X = -m m_t \psi_{(0)} + \frac{\mathrm{i}}{2} m_i \mathcal{D}_i \psi_{(0)} + \frac{\mathrm{i}}{2} \partial_i (m_i \psi_{(0)}) \,. \tag{3.22}$$

The LO equation (3.20) is defined up to the addition of any terms that by themselves transform as in (3.14). The minimal choice is to set these terms to zero.

## 3.2 NLO Schrödinger equation

The NLO equation is an equation for the NLO wave function $\psi_{(2)}$ of the form $O\psi_{(2)} + \tilde{O}\psi_{(0)}$ where $O$ and $\tilde{O}$ are operators independent of $\psi_{(0)}$ and $\psi_{(2)}$. By (3.9) we would like this to transform as

$$\begin{aligned} \delta(\text{NLO Eq.}) = {} & a^i \partial_i(\text{NLO Eq.}) - \mathrm{i} m \Lambda(\text{NLO Eq.}) \\ & + \Lambda \partial_t(\text{LO Eq.}) - \mathrm{i} m \chi(\text{LO Eq.}) + \zeta^i \partial_i(\text{LO Eq.}) \,. \end{aligned} \tag{3.23}$$

The first line corresponds to homogeneous terms and the second line to inhomogeneous terms. Adopting the same approach as for the LO equation, we can guarantee the correct transformation properties under the gauge transformations $\{\Lambda, \chi\}$ if we express the NLO equation in terms of a covariant derivative that transforms in the

same way as $\psi_{(2)}$ with respect to the $\{\Lambda, \chi\}$ transformations. By combining the transformations (2.31a)–(2.31f) with (3.9), we find that

$$
\begin{aligned}
\mathcal{D}_t \psi_{(2)} &= \partial_t \psi_{(2)} + \mathrm{i}mm_t \psi_{(2)} + \mathrm{i}mB_t \psi_{(0)} - m_t \mathcal{D}_t \psi_{(0)} \,, \\
\mathcal{D}_i \psi_{(2)} &= \partial_i \psi_{(2)} + \mathrm{i}mm_i \psi_{(2)} + \mathrm{i}m \left( B_i - \Phi_{it} \right) \psi_{(0)} - m_i \mathcal{D}_t \psi_{(0)} \,,
\end{aligned}
\tag{3.24}
$$

transform correctly, i.e.,

$$
\begin{aligned}
\delta_{\text{gauge}} \mathcal{D}_t \psi_{(2)} &= -\mathrm{i}m\Lambda \mathcal{D}_t \psi_{(2)} + \Lambda \partial_t (\mathcal{D}_t \psi_{(0)}) - \mathrm{i}m\chi \mathcal{D}_t \psi_{(0)} \,, \\
\delta_{\text{gauge}} \mathcal{D}_i \psi_{(2)} &= -\mathrm{i}m\Lambda \mathcal{D}_i \psi_{(2)} + \Lambda \partial_t (\mathcal{D}_i \psi_{(0)}) - \mathrm{i}m\chi \mathcal{D}_i \psi_{(0)} \,,
\end{aligned}
\tag{3.25}
$$

where $\delta_{\text{gauge}} = \delta_\Lambda + \delta_\chi$ denotes the combined gauge transformation. The reason we take $B_i - \Phi_{it}$ in $\mathcal{D}_i \psi_{(2)}$ is because we need the $B_i$ to ensure that we have the right transformations under $\Lambda$ and $\chi$, but $B_i$ shifts under the transformation with parameter $\tilde{\eta}_i$. However, so does $\Phi_{it}$, and therefore the difference is invariant under $\tilde{\eta}_i$. Since no other fields transform under $\tilde{\eta}_i$ this is the only way to ensure that the expressions built from these covariant derivatives will be inert under this transformation. Furthermore, since $\Phi_{it}$ is inert under both $\Lambda$ and $\chi$ we do not spoil these transformation properties of the covariant derivative. The $\tilde{\eta}_i$ transformations admit an interpretation as subleading local Lorentz boosts; more generally, the LO local Lorentz boosts are Galilean transformations with parameter $\lambda_i$ and the subleading corrections are described by $\tilde{\eta}_i$. We need all equations of motion to be invariant with respect to these LO and subleading boosts. The double spatial covariant derivative is

$$
\mathcal{D}_i \mathcal{D}_j \psi_{(2)} = \partial_i \left( \mathcal{D}_j \psi_{(2)} \right) + \mathrm{i}mm_i \mathcal{D}_j \psi_{(2)} + \mathrm{i}m \left( B_i - \Phi_{it} \right) \mathcal{D}_j \psi_{(0)} - m_i \mathcal{D}_t (\mathcal{D}_j \psi_{(0)}) \,,
\tag{3.26}
$$

which transforms as

$$
\delta_{\text{gauge}} \left( \mathcal{D}_i \mathcal{D}_j \psi_{(2)} \right) = -\mathrm{i}m\Lambda \mathcal{D}_i \mathcal{D}_j \psi_{(2)} + \Lambda \partial_t \left( \mathcal{D}_i \mathcal{D}_j \psi_{(0)} \right) - \mathrm{i}m\chi \mathcal{D}_i \mathcal{D}_j \psi_{(0)} \,.
\tag{3.27}
$$

This means that we can tentatively write the NLO equation as

$$
\text{NLO Eq.} = \mathrm{i}\mathcal{D}_t \psi_{(2)} + \frac{\tilde{k}}{2m} \mathcal{D}_i \mathcal{D}_i \psi_{(2)} + Y \,,
\tag{3.28}
$$

where $\tilde{k}$ is a real constant and where $Y$ represents any additional terms that ensure that (3.23) will hold. In order to produce the correct inhomogeneous terms in the second line of (3.23) that involve $\Lambda$ and $\chi$, we need to set $\tilde{k} = 1$. However, we will find it instructive to delay setting $\tilde{k}$ equal to unity. The $Y$ term should be inert under the $\chi$ transformation and transform under the $\Lambda$ transformation as $\delta_\Lambda Y = -\mathrm{i}m\Lambda Y$. Furthermore, the $Y$ term must be such that the whole equation transforms correctly under the $\zeta^i$ and $a^i$ transformations as well.

How do we find such an expression for $Y$ in (3.28)? If we look at (2.31a)–(2.31f), we see that the $B_t$, $B_i$ and $\Phi_{it}$ gauge fields also transform under the $\zeta^i$ gauge transformation. These are subleading diffeomorphisms. With respect to these transformations equation (3.28) transforms as

$$\delta_\zeta \left( i\mathcal{D}_t\psi_{(2)} + \frac{\tilde{k}}{2m}\mathcal{D}_i\mathcal{D}_i\psi_{(2)} + Y \right) = \zeta^j\partial_j \left( i\mathcal{D}_t\psi_{(0)} + \frac{\tilde{k}}{2m}\mathcal{D}_i\mathcal{D}_i\psi_{(0)} \right)$$

$$+ i(1-\tilde{k})\partial_t\zeta^i\mathcal{D}_i\psi_{(0)} + \frac{\tilde{k}}{2m}\left[ (\partial_i\zeta_j + \partial_j\zeta_i)\,\mathcal{D}_i\mathcal{D}_j\psi_{(0)} + \partial_i\partial_i\zeta_j\mathcal{D}_j\psi_{(0)} - im\partial_t\partial_i\zeta_i\psi_{(0)} \right]$$

$$+ \delta_\zeta Y \,. \tag{3.29}$$

If we choose $\tilde{k} = 1$ then the first term on the right hand side is equal to the LO equation (which is the last term in (3.23)) and furthermore we can get rid of the second term on the right hand side. There is thus a cancellation between terms coming from the $\mathcal{D}_t\psi_{(2)}$ term and terms coming from the $\mathcal{D}_i\mathcal{D}_i\psi_{(2)}$ term that involve $\partial_t\zeta^i$. This cancellation is important because there seems to be no terms that can be added to $Y$ that would be able to cancel a term proportional to $\partial_t\zeta^i\mathcal{D}_i\psi_{(0)}$. We wanted to highlight this, but from now on we will set $\tilde{k} = 1$. The remaining terms can be cancelled by choosing $Y$ to be

$$Y = \frac{1}{2m}\left[ -\Phi_{ij}\mathcal{D}_i\mathcal{D}_j\psi_{(0)} - \left( \partial_i\Phi_{ij} - \frac{1}{2}\partial_j\Phi_{ii} \right)\mathcal{D}_j\psi_{(0)} + im\frac{1}{2}\partial_t\Phi_{ii}\psi_{(0)} \right] + \tilde{Y}\,, \tag{3.30}$$

where $\tilde{Y}$ is inert under the $\zeta^i$ and $\chi$ gauge transformations and transforms as follows under the $\Lambda$ gauge transformations

$$\delta_\Lambda\tilde{Y} = -im\Lambda\tilde{Y}\,. \tag{3.31}$$

Since $\Phi_{ij}$ is inert under $\Lambda$ and $\chi$ gauge transformations we have

$$\delta_{\text{gauge}}Y = -im\Lambda Y\,. \tag{3.32}$$

It then follows that for this choice of $Y$ and with $\tilde{k} = 1$ the combination (3.28) transforms like in (3.23) for all gauge transformations. It is left to check that this combination also transforms correctly under the $a^i$ transformation and to fix $\tilde{Y}$.

Before we fix $\tilde{Y}$ we mention that we could have added to equation (3.26) the term $-\chi_{ij}^k\mathcal{D}_k\psi_{(2)}$ where $\chi_{ij}^k$ is given by

$$\chi_{ij}^k = \frac{1}{2}\left( \partial_i\Phi_{jk} + \partial_j\Phi_{ik} - \partial_k\Phi_{ij} \right)\,. \tag{3.33}$$

Such a term is reminiscent of a Levi–Civita connection but for the NLO diffeomorphisms generated by $\zeta_i$. The second term in parentheses in the expression for $Y$ is in fact just $\chi_{ij}^i$.

Since the $\tilde{Y}$ term is inert under $\chi$ and $\zeta^i$ and transforms covariantly under $\Lambda$ it can only be built out of covariant derivatives of $\psi_{(0)}$. Demanding that the NLO equation transforms correctly under the $a^i$ transformations we find

$$\tilde{Y} = \frac{1}{2m} M_{it} \mathcal{D}_i \psi_{(0)} - \frac{1}{2m} \mathcal{D}_t \mathcal{D}_t \psi_{(0)} + \hat{Y} \,, \tag{3.34}$$

where we defined

$$M_{\mu\nu} = 2\partial_{[\mu} m_{\nu]} \,, \tag{3.35}$$

as the field strength of $m_\mu$. This field strength also arises as the commutator of two covariant derivatives (3.16) acting on the LO wave function

$$[\mathcal{D}_\mu, \mathcal{D}_\nu]\psi_{(0)} = im M_{\mu\nu}\psi_{(0)} \,. \tag{3.36}$$

The term $\hat{Y}$ in equation (3.34) is any term that is inert under $\chi$ and $\zeta^i$ transformations and that transforms as

$$\delta\hat{Y} = a^i \partial_i \hat{Y} - im\Lambda\hat{Y} \,, \tag{3.37}$$

under the $\Lambda$ and $a^i$ transformations. The $\hat{Y}$ is not needed to make the NLO equation transform correctly and so the minimal choice is to set $\hat{Y} = 0$, which we will do in what follows. The final NLO equation is thus

$$
\begin{aligned}
\text{NLO Eq.} = {} & i\mathcal{D}_t \psi_{(2)} + \frac{1}{2m} \mathcal{D}_i \mathcal{D}_i \psi_{(2)} \\
& - \frac{1}{2m}\left[ \Phi_{ij} \mathcal{D}_i \mathcal{D}_j \psi_{(0)} + \left( \partial_i \Phi_{ij} - \frac{1}{2}\partial_j \Phi_{ii} \right) \mathcal{D}_j \psi_{(0)} \right] \\
& + i\frac{1}{4}\partial_t \Phi_{ii}\psi_{(0)} + \frac{1}{2m} M_{it} \mathcal{D}_i \psi_{(0)} - \frac{1}{2m}\mathcal{D}_t \mathcal{D}_t \psi_{(0)} \,.
\end{aligned}
\tag{3.38}
$$

The first two terms and the last term in this equation follow directly from (3.6) by replacing ordinary derivatives by covariant ones. The remaining terms then follow from covariance with respect to the $\zeta^i$ and residual $\xi^i$ transformations.

## 3.3 From Cartesian to spherical coordinates

We have chosen to work in Cartesian coordinates to keep things simple. On the other extreme one could work in an arbitrary coordinate system and study how the LO and NLO Schrödinger equations transforms under LO diffeomorphisms. This will be done in Section 3.5, but only at the level of the Lagrangian. It is often convenient to work with different LO coordinate systems, in particular spherical coordinates. The latter would arise naturally when looking at $1/c^2$ expansions of the Schwarzschild geometry in Schwarzschild coordinates. In this section we discuss how we can transform the previous Cartesian results on the LO and NLO Schrödinger equations to spherical coordinates.

Our derivation of the Schrödinger equation above involved Cartesian coordinates for the flat LO geometry. In this section, we change coordinates from spatial Cartesian coordinates $(x, y, z)$ to spherical spatial coordinates $(r, \theta, \phi)$. These coordinate systems are related by

$$
\begin{aligned}
x &= r \sin \theta \cos \phi \,, \\
y &= r \sin \theta \sin \phi \,, \\
z &= r \cos \theta \,.
\end{aligned}
\tag{3.39}
$$

Note that the absolute time $t$ is unaffected by this change of coordinates.

We denote the Cartesian coordinates by $x^1 = x$, $x^2 = y$ and $x^3 = z$ and we will denote the spherical coordinates by $x'^1 = r$, $x'^2 = \theta$ and $x'^3 = \phi$. The Cartesian metric $h_{ij}(x) = \delta_{ij}$ whereas the spherical metric $h'_{ij}(x') = \mathrm{diag}(1, r^2, r^2 \sin^2 \theta)$ such that we have

$$
\delta_{ij} dx^i dx^j = h'_{ij} dx'^i dx'^j \,.
\tag{3.40}
$$

Unlike in Cartesian coordinates, the indices $i, j, \ldots$ are raised and lowered with $h'^{ij}$ (the inverse of $h_{ij}$) and $h_{ij}$. The determinant of the metric is

$$
\sqrt{h'} = r^2 \sin \theta \,.
\tag{3.41}
$$

Changing coordinates, the Laplacian becomes

$$
\partial_i \partial_i \psi_{(0)} = \frac{1}{\sqrt{h'}} \partial'_i (\sqrt{h'} h'^{ij} \partial'_j \psi'_{(0)}) = h'^{ij} \nabla'_i \nabla'_j \psi'_{(0)} =: \Delta \psi'_{(0)} \,,
\tag{3.42}
$$

where $\psi_{(0)}(t, x, y, z) = \psi'_{(0)}(t, r, \theta, \phi)$, and where $\nabla'_i$ is the Levi-Civita connection in spherical coordinates with $\partial'_i = \frac{\partial}{\partial x'^i}$. We also have that

$$
\partial_i m_i = \frac{1}{\sqrt{h'}} \partial'_i (\sqrt{h'} m'^i) = \nabla'_i m'^i = h'^{ij} \nabla'_i m'_j \,,
\tag{3.43}
$$

where $m'_i$ obeys $m_i dx^i = m'_i dx'^i$ and where $m'^i = h'^{ij} m'_j$. This means that the LO equation with flat LO geometry in spherical coordinates becomes

$$
\text{LO Eq.} = \mathrm{i} \mathcal{D}'_t \psi'_{(0)} + \frac{1}{2m} h'^{ij} \mathcal{D}'_i \mathcal{D}'_j \psi'_{(0)} \,,
\tag{3.44}
$$

where

$$
\begin{aligned}
\mathcal{D}'_t \psi'_{(0)} &= \partial_t \psi'_{(0)} + \mathrm{i} m m'_t \psi'_{(0)} \\
\mathcal{D}'_i \psi'_{(0)} &= \partial'_i \psi'_{(0)} + \mathrm{i} m m'_i \psi'_{(0)} \\
h'^{ij} \mathcal{D}'_i \mathcal{D}'_j \psi'_{(0)} &= \Delta \psi'_{(0)} + \mathrm{i} m \psi'_{(0)} h'^{ij} \nabla'_i m'_j + 2 \mathrm{i} h'^{ij} m'_i \partial'_j \psi'_{(0)} - m^2 h'^{ij} m'_i m'_j \psi'_{(0)} \\
&= h'^{ij} \left( \nabla'_i \mathcal{D}'_j \psi'_{(0)} + \mathrm{i} m m'_i \mathcal{D}'_j \psi'_{(0)} \right) \,,
\end{aligned}
\tag{3.45}
$$

and where $m'_t(t, x') = m_t(t, x)$.

Following the same line of reasoning, we can write the NLO equation in spherical coordinates as follows

$$
\begin{aligned}
\text{NLO Eq.} = {}& \mathrm{i}\mathcal{D}'_t\psi'_{(2)} + \frac{1}{2m}h'^{ij}\mathcal{D}'_i\mathcal{D}'_j\psi'_{(2)} \\
& - \frac{1}{2m}h'^{ij}h'^{kl}\left[\Phi'_{ik}\mathcal{D}'_j\mathcal{D}'_l\psi'_{(0)} + \left(\nabla'_i\Phi'_{jk} - \frac{1}{2}\nabla'_k\Phi'_{ij}\right)\mathcal{D}'_l\psi'_{(0)}\right] \quad (3.46) \\
& + \frac{\mathrm{i}}{4}h'^{ij}\partial_t\Phi'_{ij}\psi'_{(0)} + \frac{1}{2m}h'^{ij}M'_{it}\mathcal{D}'_j\psi'_{(0)} - \frac{1}{2m}\mathcal{D}'_t\mathcal{D}'_t\psi'_{(0)},
\end{aligned}
$$

where

$$
\begin{aligned}
\mathcal{D}'_t\psi'_{(2)} &= \partial_t\psi'_{(2)} + \mathrm{i}mm'_t\psi'_{(2)} + \mathrm{i}mB'_t\psi'_{(0)} - m'_t\mathcal{D}'_t\psi'_{(0)}, \\
\mathcal{D}'_i\psi'_{(2)} &= \partial'_i\psi'_{(2)} + \mathrm{i}mm'_i\psi'_{(2)} + \mathrm{i}m\left(B'_i - \Phi'_{it}\right)\psi'_{(0)} - m'_i\mathcal{D}'_t\psi'_{(0)}, \\
\mathcal{D}'_i\mathcal{D}'_j\psi'_{(2)} &= \nabla'_i\left(\mathcal{D}'_j\psi'_{(2)}\right) + \mathrm{i}mm'_i\mathcal{D}'_j\psi'_{(2)} + \mathrm{i}m\left(B'_i - \Phi'_{it}\right)\mathcal{D}'_j\psi'_{(0)} - m'_i\mathcal{D}'_t(\mathcal{D}'_j\psi'_{(0)}),
\end{aligned}
$$
$$(3.47)$$

and where $\psi'_{(2)}(t,x') = \psi_{(2)}(t,x)$, $B'_t(t,x') = B_t(t,x)$ and $\Phi'_{ij}(t,x')dx'^i dx'^j = \Phi_{ij}(t,x)dx^i dx^j$. Of course equations (3.44) and (3.47) are valid in any coordinate system that we choose to represent a flat 3-dimensional Euclidean space such as cylindrical or oblate/prolate spherical coordinates.

## 3.4 The inner product & field redefinitions

The wave function $\Psi$ defined in (3.1) comes with a non-standard normalisation: it does not satisfy $\langle\Psi|\Psi\rangle = 1$ up to a given order in $1/c^2$; rather, this must be imposed by hand via a suitable field redefinition. The inner product $\langle\Psi|\Psi\rangle$ descends from the Klein–Gordon inner product, as we will now show. Assuming that the Lorentzian spacetime $(M,g)$, in which the Klein–Gordon theory is defined, is globally hyperbolic and stationary, the Klein–Gordon inner product of two different positive frequency solutions $\varphi_{\mathrm{KG}}$ and $\psi_{\mathrm{KG}}$ is given by (see, e.g., [53])

$$
\begin{aligned}
\langle\varphi_{\mathrm{KG}}|\psi_{\mathrm{KG}}\rangle &= \mathrm{i}c^{-1}\int_\Sigma d^dx\sqrt{\gamma}\,n^\mu\left(\psi_{\mathrm{KG}}\partial_\mu\varphi^\star_{\mathrm{KG}} - \varphi^\star_{\mathrm{KG}}\partial_\mu\psi_{\mathrm{KG}}\right) \\
&= \mathrm{i}c^{-1}\int_{t=\mathrm{cst}} d^dx\sqrt{-g}g^{\mu\nu}\partial_\nu t\left(\psi_{\mathrm{KG}}\partial_\mu\varphi^\star_{\mathrm{KG}} - \varphi^\star_{\mathrm{KG}}\partial_\mu\psi_{\mathrm{KG}}\right),
\end{aligned}
$$
$$(3.48)$$

where $\Sigma$ is a Cauchy hypersurface defined by $t = \mathrm{cst}$ with outward pointing timelike unit normal vector $n^\mu = (-g^{tt})^{-1/2}\,g^{\mu\nu}\partial_\nu t$, while $\gamma_{\mu\nu}$ is the induced metric on the hypersurface $\Sigma$ whose determinant satisfies $\sqrt{\gamma}n^\mu = \sqrt{-g}g^{\mu\nu}\partial_\nu t$. As in Eq. (3.1), we make the following decomposition of the Klein–Gordon fields

$$
\varphi_{\mathrm{KG}} = \mathrm{e}^{-\mathrm{i}mc^2 t}\Phi, \qquad \psi_{\mathrm{KG}} = \mathrm{e}^{-\mathrm{i}mc^2 t}\Psi, \qquad (3.49)
$$

which means that the inner product becomes

$$\langle\varphi_{\mathrm{KG}}|\psi_{\mathrm{KG}}\rangle = -2mc\int_{t=\mathrm{cst}}d^dx\sqrt{-g}g^{\mu\nu}\tau_\mu\tau_\nu\Psi\Phi^\star$$
$$+\,\mathrm{i}c^{-1}\int_{t=\mathrm{cst}}d^dx\sqrt{-g}g^{\mu\nu}\tau_\nu\left(\Psi\partial_\mu\Phi^\star-\Phi^\star\partial_\mu\Psi\right). \tag{3.50}$$

Using the relations

$$\sqrt{-g} = ce\left(1+c^{-2}\left(\Phi+\frac{1}{2}h^{\mu\nu}\Phi_{\mu\nu}\right)+\mathcal{O}(c^{-4})\right), \tag{3.51a}$$

$$g^{\mu\nu}\tau_\mu\tau_\nu = -c^{-2}+2c^{-4}\left(\Phi+\frac{1}{2}h^{\mu\nu}m_\mu m_\nu\right)+\mathcal{O}(c^{-6}), \tag{3.51b}$$

$$g^{\mu\nu}\tau_\nu = c^{-2}\left(v^\mu-h^{\mu\nu}m_\nu\right)+\mathcal{O}(c^{-4}), \tag{3.51c}$$

$$\Phi = \varphi_{(0)}+c^{-2}\varphi_{(2)}+\mathcal{O}(c^{-4}), \tag{3.51d}$$

$$\Psi = \psi_{(0)}+c^{-2}\psi_{(2)}+\mathcal{O}(c^{-4}), \tag{3.51e}$$

we obtain

$$\langle\varphi_{\mathrm{KG}}|\psi_{\mathrm{KG}}\rangle = 2m\int_{t=\mathrm{cst}}d^dx\,e\Big(\psi_{(0)}\varphi_{(0)}^\star+c^{-2}\Big[\psi_{(0)}\varphi_{(2)}^\star+\psi_{(2)}\varphi_{(0)}^\star+\frac{1}{2}h^{\mu\nu}\Phi_{\mu\nu}\varphi_{(0)}^\star\psi_{(0)}$$
$$+\frac{i}{2m}\left(\psi_{(0)}\left(v^\mu-h^{\mu\nu}m_\nu\right)\mathcal{D}_\mu\varphi_{(0)}^\star-\varphi_{(0)}^\star\left(v^\mu-h^{\mu\nu}m_\nu\right)\mathcal{D}_\mu\psi_{(0)}\right)\Big]+\mathcal{O}(c^{-4})\Big), \tag{3.52}$$

where $\mathcal{D}_\mu\psi_{(0)}$ is defined in equation (3.16). For a flat LO geometry in Cartesian coordinates (2.29) the inner product (3.52) becomes

$$\langle\varphi_{\mathrm{KG}}|\psi_{\mathrm{KG}}\rangle = 2m\int_{t=\mathrm{cst}}d^dx\Big[\psi_{(0)}\varphi_{(0)}^\star+c^{-2}\psi_{(0)}\Big(\varphi_{(2)}^\star-\frac{i}{2m}\mathcal{D}_t\varphi_{(0)}^\star-\frac{i}{2m}m_i\mathcal{D}_i\varphi_{(0)}^\star+\frac{1}{4}\Phi_{ii}\varphi_{(0)}^\star\Big)$$
$$+c^{-2}\varphi_{(0)}^\star\Big(\psi_{(2)}+\frac{i}{2m}\mathcal{D}_t\psi_{(0)}+\frac{i}{2m}m_i\mathcal{D}_i\psi_{(0)}+\frac{1}{4}\Phi_{ii}\psi_{(0)}\Big)+\mathcal{O}(c^{-4})\Big]. \tag{3.53}$$

We note that this inner product (3.52) is Galilean boost invariant. It is straightforward to see that the inner product is invariant under the $\chi$ transformations. To see that the inner product is invariant under the $\zeta^i$ transformations we observe that the terms at order $c^{-2}$ transform into a total derivative. We assume that the boundary terms arising from applying Stokes' theorem vanish. To see the invariance under the $\Lambda$ transformation we have to integrate by parts the transformation of the $m_i$ terms (which couple to the spatial part of the $U(1)$ Noether current) and use the LO equation of motion.

We would like the inner product to take the form

$$\langle\varphi_{\mathrm{KG}}|\psi_{\mathrm{KG}}\rangle = 2m\int_{t=\mathrm{cst}}d^dx\Big[\psi_{(0)}\varphi_{(0)}^\star+c^{-2}\left(\psi_{(0)}\hat{\varphi}_{(2)}^\star+\varphi_{(0)}^\star\hat{\psi}_{(2)}\right)+\mathcal{O}(c^{-4})\Big]$$
$$= 2m\int_{t=\mathrm{cst}}d^dx\Big(\psi_{(0)}+c^{-2}\hat{\psi}_{(2)}+\mathcal{O}(c^{-4})\Big)\Big(\varphi_{(0)}^\star+c^{-2}\hat{\varphi}_{(2)}^\star+\mathcal{O}(c^{-4})\Big). \tag{3.54}$$

This can be achieved if we define

$$\hat{\psi}_{(2)} = \psi_{(2)} + \frac{i}{2m}\mathcal{D}_t\psi_{(0)} + \frac{i}{2m}m_i\mathcal{D}_i\psi_{(0)} + \frac{1}{4}\Phi_{ii}\psi_{(0)} + iX\psi_{(0)} + X^i\partial_i\varphi_{(0)} + \frac{1}{2}\varphi_{(0)}\partial_i X^i + \dots,$$
(3.55)

where $X$ and $X^i$ are arbitrary real objects (that drop out of the inner product when integrating by parts) and where the dots denote terms proportional to the LO equation of motion. There are no obvious choices for the $X$ and $X^i$ terms that would make the redefinition simpler.

With $X^i = X = 0$, the redefinition of the NLO wave function takes the form

$$\hat{\psi}_{(2)} = \psi_{(2)} + \frac{i}{2m}\mathcal{D}_t\psi_{(0)} + \frac{i}{2m}m_i\mathcal{D}_i\psi_{(0)} + \frac{1}{4}\Phi_{ii}\psi_{(0)} =: \psi_{(2)} + \hat{O}\psi_{(0)},$$
(3.56)

where we defined the operator

$$\hat{O} = \frac{i}{2m}\mathcal{D}_t + \frac{i}{2m}m_i\mathcal{D}_i + \frac{1}{4}\Phi_{ii}.$$
(3.57)

We find that $\hat{\psi}_{(2)}$ transforms as follows under the gauge transformations $\{\Lambda, \chi, \zeta^i\}$

$$\delta_\chi\hat{\psi}_{(2)} = -im\chi\psi_{(0)},$$
(3.58a)

$$\delta_\Lambda\hat{\psi}_{(2)} = -im\Lambda\hat{\psi}_{(2)} + \Lambda\partial_t\psi_{(0)} + \frac{i}{2m}\partial_i\Lambda\mathcal{D}_i\psi_{(0)},$$
(3.58b)

$$\delta_\zeta\hat{\psi}_{(2)} = \zeta^i\partial_i\psi_{(0)} + \frac{1}{2}\psi_{(0)}\partial_i\zeta^i.$$
(3.58c)

We will next define a gauge covariant derivative $\hat{\mathcal{D}}_\mu$ that acts on $\hat{\psi}_{(2)}$ so that $\hat{\mathcal{D}}_\mu\hat{\psi}_{(2)}$ has the same transformation properties under $\Lambda$ and $\chi$ as $\hat{\psi}_{(2)}$. A convenient choice is to define the covariant derivative in the same way as we defined $\hat{\psi}_{(2)}$ in (3.56), i.e.,[4]

$$\hat{\mathcal{D}}_\mu\hat{\psi}_{(2)} := \mathcal{D}_\mu\psi_{(2)} + \hat{O}\mathcal{D}_\mu\psi_{(0)}.$$
(3.59)

More explicitly we have

$$\hat{\mathcal{D}}_t\hat{\psi}_{(2)} = \partial_t\hat{\psi}_{(2)} + imm_t\hat{\psi}_{(2)} + imB_t\psi_{(0)} - m_t\mathcal{D}_t\psi_{(0)} - \frac{i}{2m}\partial_t m_i\mathcal{D}_i\psi_{(0)}$$
$$+ \frac{1}{2}m_i M_{ti}\psi_{(0)} - \frac{1}{4}\partial_t\Phi_{ii}\psi_{(0)},$$
(3.60)

$$\hat{\mathcal{D}}_i\hat{\psi}_{(2)} = \partial_i\hat{\psi}_{(2)} + imm_i\hat{\psi}_{(2)} + im(B_i - \Phi_{it})\psi_{(0)} - m_i\mathcal{D}_t\psi_{(0)} - \frac{i}{2m}\partial_i m_j\mathcal{D}_j\psi_{(0)}$$
$$+ \frac{1}{2}m_j M_{ij}\psi_{(0)} + \frac{1}{2}M_{it}\psi_{(0)} - \frac{1}{4}\partial_i\Phi_{jj}\psi_{(0)},$$
(3.61)

---

[4]We emphasise that this choice is not unique. There are other combinations that transform correctly under $\{\Lambda, \chi\}$; for example, a more "minimal" covariant derivative, which doesn't involve $\Phi_{\mu\nu}$, is given by $\tilde{\mathcal{D}}_\mu\hat{\psi}_{(2)} = \hat{\mathcal{D}}_\mu\hat{\psi}_{(2)} + \frac{1}{4}\partial_\mu\Phi_{ii}\psi_{(0)} + \frac{i}{2m}[\mathcal{D}_\mu, \mathcal{D}_t]\psi_{(0)}$.

where it is useful to note that $[\mathcal{D}_\mu, \mathcal{D}_\nu]$ on any number of covariant derivatives acting on $\psi_{(0)}$ is equal to $\mathrm{i}mM_{\mu\nu}$ times that same set of covariant derivatives acting on $\psi_{(0)}$. It can be explicitly verified that

$$\delta_\chi \hat{\mathcal{D}}_\mu \hat{\psi}_{(2)} = \delta_\chi \mathcal{D}_\mu \psi_{(2)} = -\mathrm{i}m\chi \mathcal{D}_\mu \psi_{(0)} \,, \tag{3.62a}$$

$$\delta_\Lambda \hat{\mathcal{D}}_\mu \hat{\psi}_{(2)} = -\mathrm{i}m\Lambda \hat{\mathcal{D}}_\mu \hat{\psi}_{(2)} + \Lambda \partial_t \mathcal{D}_\mu \psi_{(0)} + \frac{\mathrm{i}}{2m} \partial_i \Lambda \mathcal{D}_i \mathcal{D}_\mu \psi_{(0)} \,. \tag{3.62b}$$

We also need an expression for the double contracted spatial covariant derivative, and we can use the same trick for this, namely we define

$$\hat{\mathcal{D}}_i \hat{\mathcal{D}}_i \hat{\psi}_{(2)} = \mathcal{D}_i \mathcal{D}_i \psi_{(2)} + \hat{O}(\mathcal{D}_i \mathcal{D}_i \psi_{(0)}) \,. \tag{3.63}$$

Explicitly, this is given by

$$\begin{aligned}
\hat{\mathcal{D}}_i \hat{\mathcal{D}}_i \hat{\psi}_{(2)} = {}& \partial_i \left( \hat{\mathcal{D}}_i \hat{\psi}_{(2)} \right) + \mathrm{i}mm_i \hat{\mathcal{D}}_i \hat{\psi}_{(2)} + \mathrm{i}m \left( B_i - \Phi_{it} \right) \mathcal{D}_i \psi_{(0)} - m_i \mathcal{D}_t \mathcal{D}_i \psi_{(0)} \\
& - \frac{\mathrm{i}}{2m} \partial_i m_j \mathcal{D}_j \mathcal{D}_i \psi_{(0)} + \frac{1}{2} m_j M_{ij} \mathcal{D}_i \psi_{(0)} + \frac{1}{2} M_{it} \mathcal{D}_i \psi_{(0)} \\
& - \frac{1}{4} \partial_i \Phi_{jj} \mathcal{D}_i \psi_{(0)} \,. 
\end{aligned} \tag{3.64}$$

This quantity transforms as

$$\delta_\chi \left( \hat{\mathcal{D}}_i \hat{\mathcal{D}}_i \hat{\psi}_{(2)} \right) = \delta_\chi \left( \mathcal{D}_i \mathcal{D}_i \psi_{(2)} \right) = -\mathrm{i}m\chi \mathcal{D}_i \mathcal{D}_i \psi_{(0)} \,, \tag{3.65a}$$

$$\delta_\Lambda \left( \hat{\mathcal{D}}_i \hat{\mathcal{D}}_i \hat{\psi}_{(2)} \right) = -\mathrm{i}m\Lambda \hat{\mathcal{D}}_i \hat{\mathcal{D}}_i \hat{\psi}_{(2)} + \Lambda \partial_t \mathcal{D}_i \mathcal{D}_i \psi_{(0)} + \frac{\mathrm{i}}{2m} \partial_i \Lambda \mathcal{D}_i \mathcal{D}_j \mathcal{D}_j \psi_{(0)} \,. \tag{3.65b}$$

We can thus recast the NLO equation (3.38) in terms of $\hat{\psi}_{(2)}$ entering the standard inner product (3.54) as

$$\begin{aligned}
\text{NLO Eq.} = {}& \mathrm{i}\hat{\mathcal{D}}_t \hat{\psi}_{(2)} + \frac{1}{2m} \hat{\mathcal{D}}_i \hat{\mathcal{D}}_i \hat{\psi}_{(2)} \\
& - \frac{1}{2m} \left[ \Phi_{ij} \mathcal{D}_i \mathcal{D}_j \psi_{(0)} + \left( \partial_i \Phi_{ij} - \frac{1}{2} \partial_j \Phi_{ii} \right) \mathcal{D}_j \psi_{(0)} \right] \\
& + \frac{\mathrm{i}}{4} \partial_t \Phi_{ii} \psi_{(0)} + \frac{1}{2m} M_{it} \mathcal{D}_i \psi_{(0)} - \frac{1}{2m} \mathcal{D}_t \mathcal{D}_t \psi_{(0)} - \hat{O}(\text{LO Eq.}) \,,
\end{aligned} \tag{3.66}$$

where we have used that the terms on which the $\hat{O}$ operator acts precisely combine to give the LO equation of motion. Thus, when imposing the LO equation of motion, the NLO equation written in terms of the redefined fields (3.56) assumes the same functional form as the NLO equation written in terms of the original fields (3.38).

As a last remark, we note that the NLO equation involving the normalised wave function (3.66) in spherical coordinates takes the form

$$
\begin{aligned}
\text{NLO Eq.} = {}& \mathrm{i}\hat{\mathcal{D}}'_t\hat{\psi}'_{(2)} + \frac{1}{2m}h'^{ij}\hat{\mathcal{D}}'_i\hat{\mathcal{D}}'_j\hat{\psi}'_{(2)} \\
& - \frac{1}{2m}h'^{ij}h'^{kl}\left[\Phi'_{ik}\mathcal{D}'_j\mathcal{D}'_l\psi'_{(0)} + \left(\nabla'_i\Phi'_{jk} - \frac{1}{2}\nabla'_k\Phi'_{ij}\right)\mathcal{D}'_l\psi'_{(0)}\right] \\
& + \frac{\mathrm{i}}{4}h'^{ij}\partial_t\Phi'_{ij}\psi'_{(0)} + \frac{1}{2m}h'^{ij}M'_{it}\mathcal{D}'_j\psi'_{(0)} - \frac{1}{2m}\mathcal{D}'_t\mathcal{D}'_t\psi'_{(0)} - \hat{O}'(\text{LO Eq.})\,,
\end{aligned}
\tag{3.67}
$$

where

$$
\begin{aligned}
\hat{\mathcal{D}}'_t\hat{\psi}'_{(2)} = {}& \partial_t\hat{\psi}'_{(2)} + \mathrm{i}mm'_t\hat{\psi}'_{(2)} + \mathrm{i}mB'_t\psi'_{(0)} - m'_t\mathcal{D}'_t\psi'_{(0)} - \frac{\mathrm{i}}{2m}h'^{ij}\partial_t m'_i\mathcal{D}'_j\psi'_{(0)} \\
& + \frac{1}{2}h'^{ij}m'_iM'_{tj}\psi'_{(0)} - \frac{1}{4}h'^{ij}\partial_t\Phi'_{ij}\psi'_{(0)}\,,
\end{aligned}
\tag{3.68}
$$

$$
\begin{aligned}
h'^{ij}\hat{\mathcal{D}}'_i\hat{\mathcal{D}}'_i\hat{\psi}'_{(2)} = {}& h'^{ij}\nabla'_i\left(\hat{\mathcal{D}}'_i\hat{\psi}'_{(2)}\right) + \mathrm{i}mh'^{ij}m'_i\hat{\mathcal{D}}'_j\hat{\psi}'_{(2)} + \mathrm{i}mh'^{ij}\left(B'_i - \Phi'_{it}\right)\mathcal{D}'_j\psi'_{(0)} \\
& - h'^{ij}m'_i\mathcal{D}'_t\mathcal{D}'_j\psi'_{(0)} - \frac{\mathrm{i}}{2m}h'^{il}h'^{jk}\nabla'_i m'_j\mathcal{D}'_k\mathcal{D}'_l\psi'_{(0)} + \frac{1}{2}h'^{ij}h'^{kl}m'_iM'_{kj}\mathcal{D}'_l\psi'_{(0)} \\
& + \frac{1}{2}h'^{ij}M'_{it}\mathcal{D}'_j\psi'_{(0)} - \frac{1}{4}h'^{ij}h'^{kl}\nabla'_i\Phi'_{kl}\mathcal{D}'_j\psi'_{(0)}\,,
\end{aligned}
\tag{3.69}
$$

and where

$$
\hat{O}' = \frac{\mathrm{i}}{2m}\mathcal{D}'_t + \frac{\mathrm{i}}{2m}h'^{ij}m'_i\mathcal{D}'_i + \frac{1}{4}h'^{ij}\Phi'_{ij}\,.
\tag{3.70}
$$

In writing the above, we used the notation introduced in Section 3.3 where a prime indicates that we are using spherical coordinates.

## 3.5 The $1/c^2$ expansion of the Klein–Gordon Lagrangian

In this section, we expand the Lagrangian for a complex scalar field in powers of $1/c^2$. This leads to an off-shell formulation of the theory we developed above. Furthermore, we will no longer restrict to a flat LO geometry, and we will see that the theory can only couple on-shell to LO geometries that admit a notion of absolute time, i.e., $\partial_{[\mu}\tau_{\nu]} = 0$ (this was also observed in [37]). For other examples of theories obtained by $1/c^2$ expansions as well as more details about the general framework of $1/c^2$ expansions, we refer to [34, 35, 37, 38, 49, 54].

The Klein–Gordon Lagrangian for a complex scalar field is

$$
\mathcal{L} = -c^{-1}\sqrt{-g}\left(g^{\mu\nu}\partial_\mu\phi_{\text{KG}}\partial_\nu\phi^\star_{\text{KG}} + m^2c^2\phi_{\text{KG}}\phi^\star_{\text{KG}}\right)\,.
\tag{3.71}
$$

Just like in (3.1), we expand the Klein–Gordon field according to

$$
\phi_{\text{KG}} = \mathrm{e}^{\mathrm{i}c^2\theta_{(0)}}\Psi\,,
\tag{3.72}
$$

and we will see that $\theta_{(0)}$ is related to absolute time. The wave function $\Psi$ admits an expansion of the form

$$\Psi = \psi_{(0)} + c^{-2}\psi_{(2)} + \mathcal{O}(c^{-4}) \,. \tag{3.73}$$

Using equations (2.2) and (2.13) for the inverse metric and the metric determinant, we can write the Lagrangian as

$$
\begin{aligned}
\mathcal{L} = &-c^4 E \Psi\Psi^\star \Pi^{\mu\nu}\partial_\mu\theta_{(0)}\partial_\nu\theta_{(0)} \\
&+ c^2 E\left[\Psi\Psi^\star(T^\mu\partial_\mu\theta_{(0)})^2 - m^2\Psi\Psi^\star + \mathrm{i}\Pi^{\mu\nu}\partial_\mu\theta_{(0)}(\Psi^\star\partial_\nu\Psi - \Psi\partial_\nu\Psi^\star)\right] \\
&+ c^0 E\left[-\Pi^{\mu\nu}\partial_\mu\Psi\partial_\nu\Psi^\star + \mathrm{i}T^\mu\partial_\mu\theta_{(0)}T^\nu(\Psi\partial_\nu\Psi^\star - \Psi^\star\partial_\nu\Psi)\right] \\
&+ c^{-2}ET^\mu T^\nu\partial_\mu\Psi\partial_\nu\Psi^\star \,.
\end{aligned}
\tag{3.74}
$$

Using furthermore the expansions (2.8) and (2.14), the Klein–Gordon Lagrangian expands as

$$\mathcal{L} = c^4\mathcal{L}_{\mathcal{O}(c^4)} + c^2\mathcal{L}_{\mathcal{O}(c^2)} + \mathcal{L}_{\mathrm{LO}} + c^{-2}\mathcal{L}_{\mathrm{NLO}} + \mathcal{O}(c^{-4}) \,, \tag{3.75}$$

where the Lagrangians at orders $c^4$ and $c^2$ are given by

$$
\begin{aligned}
\mathcal{L}_{\mathcal{O}(c^4)} = &-e\psi_{(0)}\psi_{(0)}^\star h^{\mu\nu}\partial_\mu\theta_{(0)}\partial_\nu\theta_{(0)} \,, \tag{3.76} \\
\mathcal{L}_{\mathcal{O}(c^2)} = &-e(\psi_{(0)}\psi_{(2)}^\star + \psi_{(2)}\psi_{(0)}^\star)h^{\mu\nu}\partial_\mu\theta_{(0)}\partial_\nu\theta_{(0)} - eL\psi_{(0)}\psi_{(0)}^\star h^{\mu\nu}\partial_\mu\theta_{(0)}\partial_\nu\theta_{(0)} \\
&-e\psi_{(0)}\psi_{(0)}^\star(2v^{(\mu}h^{\nu)\rho}m_\rho - h^{\mu\rho}h^{\nu\sigma}\Phi_{\rho\sigma})\partial_\mu\theta_{(0)}\partial_\nu\theta_{(0)} + e\psi_{(0)}\psi_{(0)}^\star(v^\mu\partial_\mu\theta_{(0)})^2 \\
&+eh^{\mu\nu}\partial_\mu\theta_{(0)}(\psi_{(0)}^\star\partial_\nu\psi_{(0)} - \psi_{(0)}\partial_\nu\psi_{(0)}^\star) - em^2\psi_{(0)}\psi_{(0)}^\star \,. \tag{3.77}
\end{aligned}
$$

The Lagrangian at order $c^4$ gives the equation

$$h^{\mu\nu}\partial_\nu\theta_{(0)} = 0 \,, \tag{3.78}$$

when varying[5] $\psi_{(0)}$. Upon imposing this equation, the Lagrangian $\mathcal{L}_{\mathcal{O}(c^4)}$ vanishes identically. This same equation is imposed by $\psi_{(2)}$ in $\mathcal{L}_{\mathcal{O}(c^2)}$, and combined with the equation from $\psi_{(0)}$, we find

$$v^\mu\partial_\mu\theta_{(0)} = m \,, \tag{3.79}$$

and the Lagrangian $\mathcal{L}_{\mathcal{O}(c^2)}$ again vanishes identically when imposing these equations. Together, Eqs. (3.78) and (3.79) imply that

$$\tau_\mu = -\frac{1}{m}\partial_\mu\theta_{(0)} \,. \tag{3.80}$$

This equation tells us that this theory can only be defined on backgrounds with a notion of absolute time, which in an appropriate gauge is given by $t = -\theta_{(0)}/m$.[6]

---

[5]The equation of motion for $\theta_{(0)}$ is $0 = e^{-1}\partial_\mu(e\psi_{(0)}\psi_{(0)}^\star h^{\mu\nu}\partial_\nu\theta_{(0)})$, and is identically satisfied when using the on-shell condition (3.78) for $\theta_{(0)}$.

[6]In the presence of an electromagnetic field, it is possible to relax the requirement of having an absolute time; see [37] for more details.

Going forward, we will impose this condition at the level of the Lagrangian. Had we not done so, they would have been reproduced as equations of motion for subleading components of $\Psi$. This means that the LO Lagrangian, which appears at order $c^0$, can be written as

$$\mathcal{L}_{\rm LO} = \mathrm{i}em(v^\mu - h^{\mu\nu}m_\nu)(\psi_{(0)}\partial_\mu\psi_{(0)}^\star - \psi_{(0)}^\star\partial_\mu\psi_{(0)}) - eh^{\mu\nu}\partial_\mu\psi_{(0)}\partial_\nu\psi_{(0)}^\star$$
$$-2em^2\psi_{(0)}\psi_{(0)}^\star\left(\Phi + \frac{1}{2}h^{\mu\nu}m_\mu m_\nu\right), \tag{3.81}$$

which is the Schrödinger model of [55]. We can rewrite this in terms of covariant derivatives as follows

$$\mathcal{L}_{\rm LO} = \mathrm{i}emv^\mu[\psi_{(0)}(\mathcal{D}_\mu\psi_{(0)})^\star - \psi_{(0)}^\star\mathcal{D}_\mu\psi_{(0)}] - eh^{\mu\nu}\mathcal{D}_\mu\psi_{(0)}(\mathcal{D}_\nu\psi_{(0)})^\star. \tag{3.82}$$

The equation of motion obtained by varying with respect to $\psi_{(0)}^\star$ is

$$-\mathrm{i}v^\mu\mathcal{D}_\mu\psi_{(0)} + \frac{\mathrm{i}K}{2}\psi_{(0)} = -\frac{1}{2m}e^{-1}\mathcal{D}_\mu(eh^{\mu\nu}\mathcal{D}_\nu\psi_{(0)}), \tag{3.83}$$

where

$$K = -e^{-1}\partial_\mu(ev^\mu) = -\frac{1}{2}h^{\mu\nu}\pounds_v h_{\mu\nu}, \tag{3.84}$$

with the extrinsic curvature given by $K_{\mu\nu} = -\frac{1}{2}\pounds_v h_{\mu\nu}$, where $\pounds$ denotes the Lie derivative. This extrinsic curvature is symmetric and spatial, i.e.,

$$v^\mu K_{\mu\nu} = 0. \tag{3.85}$$

When the LO geometry is flat, the LO equation of motion (3.83) reduces to (3.20) if we choose Cartesian coordinates.

Using equations (2.8), (2.10), and (2.14) for the expansions of the inverse metric and the metric determinant, we can obtain the NLO Lagrangian from (3.74) in which we set $\partial_\mu\theta_{(0)} = -m\tau_\mu$. The result can be written as

$$\mathcal{L}_{\rm NLO} = \tilde{\mathcal{L}}_{\rm NLO} + \left(\Phi + \frac{1}{2}h^{\rho\sigma}\Phi_{\rho\sigma}\right)\mathcal{L}_{\rm LO}, \tag{3.86}$$

where

$$\tilde{\mathcal{L}}_{\rm NLO} = \mathrm{i}em\left[\psi_{(0)}v^\mu\left(\mathcal{D}_\mu\psi_{(2)}\right)^\star - \psi_{(0)}^\star v^\mu\mathcal{D}_\mu\psi_{(2)} + \psi_{(2)}v^\mu(\mathcal{D}_\mu\psi_{(0)})^\star - \psi_{(2)}^\star v^\mu\mathcal{D}_\mu\psi_{(0)}\right]$$
$$- eh^{\mu\nu}\left[(\mathcal{D}_\mu\psi_{(0)})^\star\mathcal{D}_\nu\psi_{(2)} + \left(\mathcal{D}_\mu\psi_{(2)}\right)^\star\mathcal{D}_\nu\psi_{(0)}\right] \tag{3.87}$$
$$+ eh^{\mu\rho}h^{\nu\sigma}\Phi_{\rho\sigma}(\mathcal{D}_\mu\psi_{(0)})^\star\mathcal{D}_\nu\psi_{(0)} + ev^\mu v^\nu\mathcal{D}_\mu\psi_{(0)}(\mathcal{D}_\nu\psi_{(0)})^\star.$$

In writing these expressions, we used the covariant derivative with a spacetime index acting on $\psi_{(2)}$ as

$$\mathcal{D}_\mu\psi_{(2)} = \partial_\mu\psi_{(2)} + \mathrm{i}mm_\mu\psi_{(2)} + \mathrm{i}m\left(B_\mu + v^\nu\Phi_{\mu\nu}\right)\psi_{(0)} + m_\mu v^\rho\mathcal{D}_\rho\psi_{(0)}, \tag{3.88}$$

which in flat space reproduces (3.24), and where we remind the reader that $\Phi_{tt} = 0$ as follows from equation (2.18). We emphasise that we cannot just use $\tilde{\mathcal{L}}_{\text{NLO}}$ to compute the NLO equation of motion since that would miss terms arising from integrating by parts the second term in (3.86) when varying $\psi_{(0)}$ and $\psi_{(0)}^\star$.

When the LO geometry is flat, the NLO Lagrangian takes the form

$$
\begin{aligned}
\mathcal{L}_{\text{NLO}} = &- \mathrm{i}m \left[ \psi_{(0)} \left( \mathcal{D}_t \psi_{(2)} \right)^\star - \psi_{(0)}^\star \mathcal{D}_t \psi_{(2)} + \psi_{(2)} (\mathcal{D}_t \psi_{(0)})^\star - \psi_{(2)}^\star \mathcal{D}_t \psi_{(0)} \right] \\
&- (\mathcal{D}_i \psi_{(0)})^\star \mathcal{D}_i \psi_{(2)} - \left( \mathcal{D}_i \psi_{(2)} \right)^\star \mathcal{D}_i \psi_{(0)} + \mathcal{D}_t \psi_{(0)} (\mathcal{D}_t \psi_{(0)})^\star \\
&+ \Phi_{ij} \left( \mathcal{D}_i \psi_{(0)} \right)^\star \mathcal{D}_j \psi_{(0)} \\
&+ \left( \Phi + \frac{1}{2} \Phi_{jj} \right) \left( -\mathrm{i}m(\psi_{(0)}(\mathcal{D}_t \psi_{(0)})^\star - \psi_{(0)}^\star \mathcal{D}_t \psi_{(0)}) - \mathcal{D}_i \psi_{(0)} (\mathcal{D}_i \psi_{(0)})^\star \right) ,
\end{aligned}
\tag{3.89}
$$

and the variation with respect to $\psi_{(0)}^\star$ produces the NLO equation plus a contribution proportional to the LO equation of motion[7]

$$
0 = \left( \Phi + \frac{1}{2} \Phi_{jj} \right) \left[ \mathrm{i}\mathcal{D}_t \psi_{(0)} + \frac{1}{2m} \mathcal{D}_i \mathcal{D}_i \psi_{(0)} \right] + \text{NLO Eq.} ,
\tag{3.90}
$$

where NLO Eq. is given in (3.38).

The final step in our derivation of the NLO Schrödinger equation involves identifying the normalised wave function as explained in Section 3.4. Generalising the result (3.56) to curved space is straightforward and produces the relation

$$
\hat{\psi}_{(2)} = \psi_{(2)} - \frac{\mathrm{i}}{2m} \hat{v}^\mu \mathcal{D}_\mu \psi_{(0)} + \frac{1}{4} h^{\mu\nu} \Phi_{\mu\nu} \psi_{(0)} =: \psi_{(2)} + \hat{O} \psi_{(0)} ,
\tag{3.91}
$$

with

$$
\hat{O} = -\frac{\mathrm{i}}{2m} \hat{v}^\mu \mathcal{D}_\mu + \frac{1}{4} h^{\mu\nu} \Phi_{\mu\nu} .
\tag{3.92}
$$

This step can be implemented at the level of the NLO Lagrangian, which we discuss further in Appendix A.

# 4  Quantum mechanics in a Kerr background

In this section, we study the Schrödinger equation on a nonrelativistic approximation of the Kerr background. By expanding the Kerr metric in powers of $1/c^2$, we obtain a "generalised" Lense–Thirring metric that is valid beyond the regime of slow rotations. By identifying the appropriate nonrelativistic geometry, we can apply the formalism developed in Section 3 to write down the LO and NLO Schrödinger equation on this background.

---

[7]The LO equation arises as the equation of motion for $\psi_{(2)}^\star$ in the NLO Lagrangian.

### 4.1 From Kerr to Lense–Thirring

The Kerr metric in Boyer–Lindquist coordinates can be thought of as a deformation of Minkowski spacetime written in terms of oblate spherical coordinates

$$x = \sqrt{R^2 + a^2}\sin\Theta\cos\phi\,, \tag{4.1a}$$

$$y = \sqrt{R^2 + a^2}\sin\Theta\sin\phi\,, \tag{4.1b}$$

$$z = R\cos\Theta\,, \tag{4.1c}$$

where $a$ is a fixed length. The flat metric in oblate spherical coordinates is

$$ds^2_{\text{flat}} = -c^2 dt^2 + \frac{\Sigma}{R^2 + a^2}dR^2 + \Sigma d\Theta^2 + (R^2 + a^2)\sin^2\Theta\, d\phi^2\,, \tag{4.2}$$

where

$$\Sigma = R^2 + a^2\cos^2\Theta\,. \tag{4.3}$$

We can write the Kerr metric as

$$ds^2_{\text{Kerr}} = ds^2_{\text{flat}} + \frac{\Sigma r_s R}{\Delta(R^2 + a^2)}dR^2 + \frac{r_s R}{\Sigma}\left(-cdt + a\sin^2\Theta\, d\phi\right)^2\,, \tag{4.4}$$

where

$$r_s = \frac{2GM}{c^2}\,, \tag{4.5a}$$

$$a = \frac{J}{cM}\,, \tag{4.5b}$$

$$\Delta = R^2 + a^2 - r_s R\,, \tag{4.5c}$$

where $J$ and $M$ are the angular momentum and mass, respectively, of the Kerr background. We assume $J$ and $M$ to be independent of $c$ (see [44] for alternative choices). This implies that the metric can be expanded in $c^{-2}$, i.e., without using odd powers of $c^{-1}$. The definition of the oblate spherical coordinates (4.1) implies the following relation[8]

$$\frac{x^2 + y^2}{R^2 + a^2} + \frac{z^2}{R^2} = 1\,. \tag{4.7}$$

Hence, surfaces in $\mathbb{R}^3$ of constant $R$ form oblate spheroids. The solution to this equation has a $1/c^2$ expansion of the form

$$R^2 = r^2 - \frac{J^2}{M^2}\frac{x^2 + y^2}{r^2 c^2} + \mathcal{O}(c^{-4})\,, \tag{4.8}$$

---

[8]It also implies that

$$\frac{x^2 + y^2}{\sin^2\Theta} - \frac{z^2}{\cos^2\Theta} = a^2\,, \tag{4.6}$$

which shows that surfaces of constant $\Theta$ are hyperboloids of revolution. For $a = 0$ this becomes the equation of a cone.

where $r^2 = x^2 + y^2 + z^2$ is the radial coordinate of a spherical coordinate system. The relation between the Cartesian coordinate $z$ and the spherical oblate angular coordinate $\Theta$ in (4.1c) combined with the expansion of $R$ in (4.8) tells us that

$$\Theta = \cos^{-1}\left(\frac{z}{R}\right) = \cos^{-1}\left(\frac{z}{r} + \mathcal{O}(c^{-2})\right) = \cos^{-1}\left(\frac{z}{r}\right) + \mathcal{O}(c^{-2}) = \theta + \mathcal{O}(c^{-2}), \quad (4.9)$$

where $\theta$ is the polar angle coordinate of a spherical coordinate system. This means that we can also write the expansion (4.8) as

$$R^2 = r^2 - \frac{J^2}{M^2}\frac{\sin^2\theta}{c^2} + \mathcal{O}(c^{-4}). \quad (4.10)$$

Making factors of $c$ explicit in (4.4) and expanding to order $c^{-2}$, using our results above, we get

$$ds^2_{\text{Kerr}} = -\left(1 - \frac{2GM}{rc^2} + \frac{2GJ^2}{Mr^3c^4}P_2(\cos\theta)\right)c^2dt^2 + \left(1 + \frac{2GM}{rc^2}\right)dr^2$$
$$+ r^2d\theta^2 + r^2\sin^2\theta\,d\phi^2 - \frac{4GJ}{rc^2}\sin^2\theta\,dtd\phi + \mathcal{O}(c^{-4}), \quad (4.11)$$

where

$$P_2(x) = \frac{1}{2}(3x^2 - 1), \quad (4.12)$$

is the second order Legendre polynomial. We remark that this metric is the $1/c$ expansion of the Hartle–Thorne metric [56] specified to the case of the Kerr BH. It would be interesting to consider $1/c$ expansions of the general Hartle–Thorne metric which is an approximate solution outside a rotating object. In addition to mass and angular momentum, the Hartle–Thorne metric contains a quadrupole moment as a free parameter.

If we assume that the black hole rotates slowly, $J \ll 1$, so that we can ignore the $J^2$ term, the above reduces to the Lense–Thirring metric. Setting $J = 0$ lands us on the $1/c^2$ expansion of the Schwarzschild metric, which we consider in Section A.2. The geometric data of the nonrelativistic geometry is given by

$$\begin{aligned}
\tau_\mu dx^\mu &= dt, \\
h_{\mu\nu}dx^\mu dx^\nu &= dr^2 + r^2d\theta^2 + r^2\sin^2\theta d\phi^2, \\
m_\mu dx^\mu &= -\frac{GM}{r}dt, \\
\Phi_{\mu\nu}dx^\mu dx^\nu &= \frac{2GM}{r}dr^2, \\
B_\mu dx^\mu &= \frac{GJ^2}{Mr^3}P_2(\cos\theta)\,dt + \frac{2GJ}{r}\sin^2\theta\,d\phi - \frac{G^2M^2}{2r^2}dt.
\end{aligned} \quad (4.13)$$

Thus, the LO geometry is flat in spherical coordinates (cf., Section 3.3). All the rotational aspects (terms proportional to $J$) are captured by the $B_\mu$ gauge field. This means that the LO Schrödinger equations does not notice the rotation.

## 4.2 The LO and NLO Schrödinger equation on a Kerr background

On a Kerr background, where the geometric structure up to NLO is given by (4.13), the LO equation of motion (3.44) becomes

$$i\partial_t \psi_{(0)} = -\frac{1}{2m}\Delta\psi_{(0)} - G\frac{mM}{r}\psi_{(0)}\,. \tag{4.14}$$

The NLO equation (3.67) becomes

$$
\begin{aligned}
i\partial_t \hat{\psi}_{(2)} = &-\frac{1}{2m}\Delta\hat{\psi}_{(2)} - \frac{GmM}{r}\hat{\psi}_{(2)} + \frac{GM}{mr}\partial_r^2\psi_{(0)} \\
&+ \frac{mGJ^2}{Mr^3}P_2(\cos\theta)\psi_{(0)} - \frac{mG^2M^2}{2r^2}\psi_{(0)} - 2iGJr^{-3}\partial_\phi\psi_{(0)} \\
&- \frac{GM}{r}i\mathcal{D}_t\psi_{(0)} + \frac{1}{2m}\mathcal{D}_t\mathcal{D}_t\psi_{(0)} + \frac{GM}{2m}\psi_{(0)}\Delta\left(r^{-1}\right)
\end{aligned} \tag{4.15}
$$

where we dropped the prime we used earlier for fields in spherical coordinates. Using the LO equation of motion (4.14), we can write the double covariant time derivative as

$$\frac{1}{2m}\mathcal{D}_t\mathcal{D}_t\psi_{(0)} = -\frac{1}{8m^3}\Delta^2\psi_{(0)} - \frac{GM}{4m}\psi_{(0)}\Delta(r^{-1}) + \frac{GM}{2mr^2}\partial_r\psi_{(0)}\,. \tag{4.16}$$

Hence, by using the LO equation (4.14), we can write the NLO equation as

$$
\begin{aligned}
i\partial_t \hat{\psi}_{(2)} = &-\frac{1}{2m}\Delta\hat{\psi}_{(2)} - \frac{GmM}{r}\hat{\psi}_{(2)} + \frac{mGJ^2 P_2(\cos\theta)}{Mr^3}\psi_{(0)} \\
&- \frac{mG^2M^2}{2r^2}\psi_{(0)} - \frac{2GJ}{r^3}i\partial_\phi\psi_{(0)} - \frac{1}{8m^3}\Delta^2\psi_{(0)} \\
&+ \frac{GM}{4m}\Delta(r^{-1})\psi_{(0)} + \frac{GM}{rm}\partial_r^2\psi_{(0)} + \frac{GM}{2mr^2}\partial_r\psi_{(0)} + \frac{GM}{2mr}\Delta\psi_{(0)}\,.
\end{aligned} \tag{4.17}
$$

This allows us to read off the effective Hamiltonian governing the evolution of the wave function in a "generalised Lense–Thirring" background. We define the total NLO wave function to be

$$\hat{\Psi} = \psi_{(0)} + c^{-2}\hat{\psi}_{(2)}\,, \tag{4.18}$$

and adding the LO and NLO equations as

$$\text{LO Eq.} + c^{-2}\text{NLO Eq.}\,, \tag{4.19}$$

we find that

$$i\partial_t \hat{\Psi} = H\hat{\Psi} + \mathcal{O}(c^{-4})\,, \tag{4.20}$$

with

$$
\begin{aligned}
H = &-\frac{1}{2m}\Delta - \frac{GmM}{r} + \frac{GM}{4mc^2}\Delta(r^{-1}) + \frac{mGJ^2 P_2(\cos\theta)}{Mc^2 r^3} \\
&- \frac{mG^2M^2}{2c^2 r^2} - \frac{2GJ}{c^2 r^3}i\partial_\phi - \frac{1}{8c^2 m^3}\Delta^2 \\
&+ \frac{GM}{c^2 m}\left(\frac{1}{r}\partial_r^2 + \frac{1}{2r^2}\partial_r + \frac{1}{2r}\Delta\right)
\end{aligned} \tag{4.21}
$$

the Hamiltonian.

The inner product is

$$\langle\phi|\psi\rangle = \int dr d\theta d\phi\, r^2 \sin\theta\, \phi^\star\psi \,. \tag{4.22}$$

An operator $O$ is Hermitian if $\langle\phi|O\psi\rangle = \langle O\phi|\psi\rangle$ for all $\phi,\psi$. Any $O$ of the form

$$O = f(r)\partial_r^2 + \left(\partial_r f + \frac{2}{r}f\right)\partial_r \,, \tag{4.23}$$

where $f$ is any radial function is Hermitian (ignoring issues with boundary terms or fall off conditions for the fields $\psi,\psi$). The Laplacian in spherical coordinates is

$$\Delta = \partial_r^2 + \frac{2}{r}\partial_r + \frac{1}{r^2}\Delta_{S^2} \,, \tag{4.24}$$

where $\Delta_{S^2}$ is the Laplacian on the unit 2-sphere.

Using the above we can see that the radial part of the last three terms in (4.21) conspire to form the Hermitian combination

$$\frac{3}{2r}\left(\partial_r^2 + \frac{1}{r}\partial_r\right) \,. \tag{4.25}$$

We thus have

$$\begin{aligned}
H = &-\frac{1}{2m}\Delta - \frac{GmM}{r} + \frac{GM}{4mc^2}\Delta(r^{-1}) + \frac{mGJ^2 P_2(\cos\theta)}{Mc^2 r^3} \\
&-\frac{mG^2 M^2}{2c^2 r^2} - \frac{2GJ}{c^2 r^3}\mathrm{i}\partial_\phi - \frac{1}{8c^2 m^3}\Delta^2 \\
&+\frac{GM}{c^2 m}\left(\frac{3}{2r}\left(\partial_r^2 + \frac{1}{r}\partial_r\right) + \frac{1}{2r^3}\Delta_{S^2}\right) \,.
\end{aligned} \tag{4.26}$$

If we denote by $\vec{L}$ the Hermitian angular momentum operator with components $L_x, L_y, L_z$ then we have

$$L_z = -\mathrm{i}\partial_\phi \,, \qquad \Delta_{S^2} = L^2 = \vec{L}\cdot\vec{L} \,. \tag{4.27}$$

Finally, if we define, as usual, the momentum $p_i = -\mathrm{i}\partial_i$ in Cartesian coordinates then we get

$$\begin{aligned}
H = &\frac{p^2}{2m} - \frac{GmM}{r} + \frac{GM}{4mc^2}\Delta(r^{-1}) - \frac{p^4}{8c^2 m^3} \\
&+\frac{GM}{2mc^2 r^3}\left(-3x^i p_i x^j p_j + L^2\right) - \frac{mG^2 M^2}{2c^2 r^2} \\
&+\frac{2GJ}{c^2 r^3}L_z + \frac{mGJ^2 P_2(\cos\theta)}{Mc^2 r^3} \,,
\end{aligned} \tag{4.28}$$

where we wrote $r^{-1}\left(\partial_r^2 + \frac{1}{r}\partial_r\right) = r^{-3} r\partial_r(r\partial_r)$ as $-r^{-3}x^i p_i x^j p_j$. This is our final result for the Hamiltonian of a spinless particle in a Kerr background up to order $c^{-2}$. The $JL_z$ coupling in the Hamiltonian has also appeared in, e.g., Refs. [57–59] which consider the Lense–Thirring effect in quantum mechanics. The result above includes further novel effects of order $J^2$ which may potentially be measurable.

# 5   Discussion & outlook

We conclude with a discussion of our results and an overview of future directions. We have presented a general framework based on symmetries for deriving the Schrödinger equation on a given gravitational background that can in principle be applied at any order in $1/c^2$ and for a wide range of metrics. This relied on recent advances in the description of nonrelativistic geometry using covariant $1/c^2$ expansions, which allowed us to use symmetry to uniquely fix the form of the equations. Complimentary to a "bottom-up" perspective, we showed that it is also possible to get these equations by $1/c^2$ expanding the Klein–Gordon Lagrangian. We then used this formalism to write down the Schrödinger equation on a Kerr background up to $\mathcal{O}(c^{-2})$, which led to a generalised Lense–Thirring geometry and to a novel Hamiltonian on this geometry.

With the ultimate goal of deriving a general minimal coupling prescription that allows us to write down the Schrödinger equation on a post-Newtonian geometry at arbitrary order in $1/c$, this work paves the way for many interesting avenues of research. A general minimal coupling prescription should, in particular, make it immediately clear what the covariant derivatives at any given order are, and so likely requires us to understand better the representation theory of the $1/c^2$ expansion of the Poincaré algebra. An immediate generalisation of the methods we develop would be to include odd powers; i.e., to consider a $1/c$ expansion rather than a $1/c^2$ expansion. This will lead to a different geometric structure compared to Section 2, and would allow for the expansion of a much more general class of metrics, including metrics in Kerr–Schild form and metrics that include retardation effects such as pp waves. Moreover, the inclusion of electromagnetism and spin would allow us to apply our formalism to a much broader class of physical systems.

In this work, we have only discussed single particles. It would be very interesting to extend the formalism to describe composite systems, electromagnetic interactions and spin. It was recently shown that when going beyond particles that are fully described by a single parameter $m$, new effects can arise. For example, systems that keep track of time, and are in quantum superpositions that are delocalised over a region in the gravitational field, will experience time-dilation induced entanglement between the internal and external degrees of freedom [9, 21]. This can result in new effects that can be probed in experiments, such as decoherence of superpositions or dephasing of clocks [9, 11, 25]. Importantly, such effects only arise within the quantum framework when post-Newtonian corrections are included, and thus their observation amounts to a test of GR in an entirely new domain. A general geometric formalism that includes such effects will be able to highlight what aspects of the theory are probed and how to design novel tests that go beyond the current paradigms. The methods presented in this paper are ideally suited to isolate fundamental principles that can become accessible in such experiments, and to pave the way for novel

experimental designs to probe the elusive interplay between quantum systems and general relativity.

## Acknowledgements

The work of JH is supported by the Royal Society University Research Fellowship Renewal "Non-Lorentzian String Theory" (grant number URF\R\221038). The work of EH is supported by Jelle Hartong's Royal Society University Research Fellowship via an enhancement award. IP was supported by the European Research Council under grant no. 742104, the Swedish Research Council under grants nos. 2019-05615 and 335-2014-7424, and the Branco Weiss Fellowship – Society in Science. The work of NO is supported in part by VR project grant 2021-04013 and Villum Foundation Experiment project 00050317. Nordita is supported in part by Nordforsk.

## A Schrödinger's equation on Schwarzschild backgrounds

In this appendix, we perform the $1/c^2$ expansion of the Schwarzschild metric in isotropic coordinates and use the methods of Section 3 to write down the Schrödinger equation, which we then match with the results of [30].

### A.1 Schwarzschild metric in isotropic coordiantes

In four dimensions, the Schwarzschild metric in isotropic coordinates was first written down by Eddington [60] and reads

$$ds^2 = -\frac{\left(1 - \frac{GM}{2rc^2}\right)^2}{\left(1 + \frac{GM}{2rc^2}\right)^2}c^2dt^2 + \left(1 + \frac{GM}{2rc^2}\right)^4 \delta_{ij}dx^i dx^j \,, \tag{A.1}$$

where $r^2 = x^2 + y^2 + z^2$. Expanding this to $\mathcal{O}(c^{-2})$, we get

$$ds^2 = -\left(1 - \frac{2GM}{rc^2} + \frac{2G^2M^2}{r^2c^4}\right)c^2dt^2 + \left(1 + \frac{2GM}{rc^2}\right)\delta_{ij}dx^i dx^j + \mathcal{O}(c^{-4})\,. \tag{A.2}$$

This allows us to read off the fields that describe the geometric data

$$\tau_\mu dx^\mu = dt \,, \tag{A.3a}$$

$$m_\mu dx^\mu = \Phi dt \,, \tag{A.3b}$$

$$h_{\mu\nu}dx^\mu dx^\nu = \delta_{ij}dx^i dx^j \,, \tag{A.3c}$$

$$\Phi_{\mu\nu}dx^\mu dx^\nu = -2\Phi\delta_{ij}dx^i dx^j \,, \tag{A.3d}$$

$$B_\mu dx^\mu = \frac{1}{2}\Phi^2 dt \,. \tag{A.3e}$$

where, for simplicity, we defined

$$\Phi := -\frac{GM}{r}\,. \tag{A.4}$$

The LO equation (3.20) is

$$\mathrm{i}\mathcal{D}_t\psi_{(0)} = -\frac{1}{2m}\partial^2\psi_{(0)}\,, \tag{A.5}$$

which we can also explicitly write as

$$\mathrm{i}\partial_t\psi_{(0)} = -\frac{1}{2m}\partial^2\psi_{(0)} + m\Phi\psi_{(0)} = -\frac{1}{2m}\partial^2\psi_{(0)} - \frac{GmM}{r}\psi_{(0)}\,. \tag{A.6}$$

The NLO equation (3.38) on this background becomes

$$\mathrm{i}\mathcal{D}_t\psi_{(2)} = -\frac{1}{2m}\partial^2\psi_{(2)} + \frac{1}{2m}\mathcal{D}_t\mathcal{D}_t\psi_{(0)} - \frac{\Phi}{m}\partial^2\psi_{(0)}\,, \tag{A.7}$$

where

$$\mathcal{D}_t\psi_{(2)} = \partial_t\psi_{(0)} + \mathrm{i}m\Phi\psi_{(2)} + \frac{\mathrm{i}}{2}m\Phi^2\psi_{(0)} - \Phi\mathcal{D}_t\psi_{(0)}\,. \tag{A.8}$$

Rewriting the NLO equation (A.7) using the LO equation (A.5), we obtain

$$\begin{aligned}
\mathrm{i}\partial_t\psi_{(2)} = &-\frac{1}{2m}\partial^2\psi_{(2)} + m\Phi\psi_{(2)} - \frac{3\Phi}{2m}\partial^2\psi_{(0)} - \frac{1}{8m^3}\partial^4\psi_{(0)} \\
&+ \frac{m}{2}\Phi^2\psi_{(0)} + \frac{\partial^2\Phi}{4m}\psi_{(0)} + \frac{1}{2m}\partial_i\Phi\partial_i\psi_{(0)}\,,
\end{aligned} \tag{A.9}$$

where we used that

$$\frac{1}{2m}\mathcal{D}_t\mathcal{D}_t\psi_{(0)} = -\frac{1}{8m^3}\partial^4\psi_{(0)} + \frac{1}{4m}(\partial^2\Phi)\psi_{(0)} + \frac{1}{2m}\partial_i\Phi\partial_i\psi_{(0)}\,. \tag{A.10}$$

We can also write this in terms of the wave function $\hat{\psi}_{(2)}$. For Schwarzschild in isotropic coordinates, the operator $\hat{O}$ given in (3.92) becomes

$$\hat{O} = \frac{\mathrm{i}}{2m}\mathcal{D}_t - \frac{3}{2}\Phi = \frac{\mathrm{i}}{2m}\partial_t\psi_{(0)} - 2\Phi\psi_{(0)}\,. \tag{A.11}$$

Hence, the NLO equation written in terms of the wavefunction $\hat{\psi}_{(2)}$ in (3.66) takes the form

$$\begin{aligned}
\mathrm{i}\partial_t\hat{\psi}_{(2)} = &-\frac{1}{2m}\partial^2\hat{\psi}_{(2)} + m\Phi\hat{\psi}_{(2)} + \frac{1}{2m}\mathcal{D}_t\mathcal{D}_t\psi_{(0)} + \mathrm{i}\Phi\mathcal{D}_t\psi_{(0)} \\
&+ \frac{m\Phi^2}{2}\psi_{(0)} - \frac{\partial^2\Phi}{m}\psi_{(0)} - \frac{\Phi}{m}\partial^2\psi_{(0)} - \frac{2}{m}\partial_i\Phi\partial_i\psi_{(0)} + \hat{O}\,(\text{LO Eq.})\,,
\end{aligned} \tag{A.12}$$

and imposing the LO equation produces the equation

$$\begin{aligned}
\mathrm{i}\partial_t\hat{\psi}_{(2)} = &-\frac{1}{2m}\partial^2\hat{\psi}_{(2)} + m\Phi\hat{\psi}_{(2)} - \frac{1}{8m^3}\partial^4\psi_{(0)} \\
&+ \frac{m\Phi^2}{2}\psi_{(0)} - \frac{3\partial^2\Phi}{4m}\psi_{(0)} - \frac{3\Phi}{2m}\partial^2\psi_{(0)} - \frac{3}{2m}\partial_i\Phi\partial_i\psi_{(0)}\,.
\end{aligned} \tag{A.13}$$

Defining $\Psi = \psi_{(0)} + c^{-2}\psi_{(2)}$ and adding the LO and NLO equations as

$$\text{LO Eq.} + c^{-2}\text{NLO Eq.}, \tag{A.14}$$

we find that the equations of motion above can be combined to give

$$
\begin{aligned}
i\partial_t \Psi = &-\frac{1}{2m}\partial^2\Psi + m\Phi\Psi - \frac{3\Phi}{2mc^2}\partial^2\Psi - \frac{1}{8m^3c^2}\partial^4\Psi \\
&+ \frac{m}{2c^2}\Phi^2\Psi + \frac{\partial^2\Phi}{4mc^2}\Psi + \frac{1}{2mc^2}\partial_i\Phi\partial_i\Psi.
\end{aligned}
\tag{A.15}
$$

In [30], Lämmerzahl writes down the $1/c^2$ expansion of the Klein–Gordon equation on a background given by the parameterised post–Newtonian (PPN) metric. The dictionary between Schwarzschild in isotropic coordinates and the PPN metric is

$$\beta = \gamma = 1, \qquad U = -\Phi, \tag{A.16}$$

in which case (A.15) matches Eq. (8) of [30].

Doing the same for (A.13) and defining $\hat{\Psi} = \psi_{(0)} + c^{-2}\hat{\psi}_{(2)}$ gives

$$
\begin{aligned}
i\partial_t \hat{\Psi} = &-\frac{1}{2m}\partial^2\hat{\Psi} + m\Phi\hat{\Psi} - \frac{1}{8c^2m^3}\partial^4\hat{\Psi} \\
&+ \frac{m\Phi^2}{2c^2}\hat{\Psi} - \frac{3\partial^2\Phi}{4mc^2}\hat{\Psi} - \frac{3\Phi}{2mc^2}\partial^2\hat{\Psi} - \frac{3}{2mc^2}\partial_i\Phi\partial_i\hat{\Psi} + \mathcal{O}(c^{-4}),
\end{aligned}
\tag{A.17}
$$

which agrees with Eq. (16) in [30].

## A.2 From Kerr to Schwarzschild

At this stage, one may wonder how to obtain (A.17) from the results of Section 4, where we expanded the Kerr metric in Boyer–Lindquist coordinates. Setting $a = 0$ in the expression for the Kerr metric (4.4) gives the Schwarzschild metric in Schwarzschild coordinates, which are related to the isotropic coordinates employed in Section A.1 by a $c$-dependent coordinate transformations. In general, $c$-dependent coordinate transformations mix LO terms with NLO terms. Moreover, although the wave function $\Psi$ that descends directly from the Klein–Gordon field, cf., (3.1), transforms as a scalar under $c$-dependent coordinate transformations, the same is *not* true for the normalised wave function $\hat{\Psi}$. This is because the operator $\hat{O}$ defined in (3.92) receives $1/c^2$ corrections. The fact that $\hat{\Psi}$ transforms differently under $c$-dependent coordinate transformations is already evident from the different transformations of $\psi_{(2)}$ and $\hat{\psi}_{(2)}$ under infinitesimal subleading diffeomorphisms $\zeta$ in (3.9) and (3.58c), respectively. In this appendix, we illustrate this using two well-known coordinate systems for the Schwarzschild metric: Schwarzschild coordinates and isotropic coordinates, and ultimately show how this allows us to match with our results for Kerr.

### A.2.1 Schwarzschild coordinates

The Schwarzschild metric in Schwarzschild coordinates $(t, r, \theta, \phi)$ is

$$ds^2_{\text{Schw}} = -\left(1 - \frac{2GM}{c^2 r}\right)c^2 dt^2 + \left(1 - \frac{2GM}{c^2 r}\right)^{-1} dr^2 + r^2 ds^2_{S^2}\,, \qquad (\text{A.18})$$

where $ds^2_{S^2} = d\theta^2 + \sin^2(\theta)d\phi^2$ is the metric on the unit two-sphere and $M$ is the mass of the black hole. Setting $J = 0$ in (4.13) gives gives the following geometric data up to $1/c^2$

$$
\begin{aligned}
\tau_\mu dx^\mu &= dt\,, \\
h_{\mu\nu}dx^\mu dx^\nu &= dr^2 + r^2 d\theta^2 + r^2 \sin^2\theta d\phi^2\,, \\
m_\mu dx^\mu &= -\frac{GM}{r}dt\,, \\
\Phi_{\mu\nu}dx^\mu dx^\nu &= \frac{2GM}{r}dr^2\,, \\
B_\mu dx^\mu &= -\frac{G^2 M^2}{2r^2}dt\,.
\end{aligned}
\qquad (\text{A.19})
$$

The Schwarzschild coordinates are related to the isotropic coordinates used in Appendix A, which we here denote by $(t, r', \theta, \phi)$ or, in Cartesian form, $(t, x, y, z)$ by the $c$-dependent coordinate transformation [60]

$$r = \left(1 + \frac{GM}{2c^2 r'}\right)^2 r'\,, \qquad (\text{A.20})$$

where

$$r'^2 = x^2 + y^2 + z^2\,, \qquad (\text{A.21})$$

with

$$
\begin{aligned}
x &= r' \sin\theta \cos\phi\,, \\
y &= r' \sin\theta \sin\phi\,, \\
z &= r' \cos\theta\,.
\end{aligned}
\qquad (\text{A.22})
$$

Note in particular that time $t$ and the angles $(\theta, \phi)$ in Schwarzschild coordinates are the same as the time and angles in isotropic coordinates when expressed in spherical form.

### A.2.2 Schrödinger's Lagrangian on Schwarzschild backgrounds

The comparison between the NLO theory in Schwarzschild (which are unprimed) and isotropic (which carry a prime) coordinates is perhaps most transparent at the level of the Lagrangians that we worked out in Section 3.5. When going from Schwarzschild coordinates to isotropic coordinates, the $c$-dependent rescaling of the radial direction (A.20) implies that the NLO Lagrangian in isotropic coordinates $\mathcal{L}'_{\text{Iso}}$ picks up

a contribution from the LO Lagrangian (cf., the expansion (3.75)). While the wave function $\Psi$ defined in (3.1) is a scalar, and as such transforms as $\Psi'(r') = \Psi(r)$ (omitting the remaining coordinates),i.e.,

$$\psi_{(0)}(r) = \psi'_{(0)}(r') \qquad \text{and} \qquad \psi_{(2)}(r) = \psi'_{(2)}(r') \,, \tag{A.23}$$

the same is *not* true for the normalised wave functions that we discussed in Section 3.4. Defining

$$\hat{\psi}_{\text{Schw}(2)} := \psi_{(2)} + \hat{O}_{\text{Schw}}\psi_{(0)} \qquad \text{and} \qquad \hat{\psi}'_{\text{Iso}(2)} := \psi'_{(2)} + \hat{O}'_{\text{Iso}}\psi'_{(0)} \,, \tag{A.24}$$

the expansion of the inner product in (3.52) that led to the definition of $\hat{O}$ implies that when changing coordinates from Schwarzschild to isotropic, $O'_{\text{Iso}}$ will pick up $1/c^2$ corrections. In other words, $\hat{O}$, and by extension $\hat{\Psi}$, do not transform as scalars usually do, i.e., as in (A.23). Taking this into account results in a commutative diagram of NLO theories

$$
\begin{array}{ccc}
\mathcal{L}_{\text{Schw,NLO}} & \xrightarrow{\;\prime\;} & \mathcal{L}_{\text{Iso,NLO}} \\
\downarrow{\scriptstyle\hat{O}_{\text{Schw}}} & & \downarrow{\scriptstyle\hat{O}'_{\text{Iso}}} \\
\hat{\mathcal{L}}_{\text{Schw,NLO}} & \xrightarrow{\;\prime\;} & \hat{\mathcal{L}}_{\text{Iso,NLO}}
\end{array}
$$

Lagrangians decorated with a hat are expressed in terms of the normalised wave functions in (A.24) which are implemented by the operator $\hat{O}$ in (3.92), while the prime denotes changing coordinates from Schwarzschild to isotropic. In what follows, we will explicitly demonstrate that this diagram commutes.

Starting with Schwarzschild coordinates, the relevant derivatives that appear in the NLO Lagrangian (3.86) are

$$
\begin{aligned}
\mathcal{D}_t\psi_{(2)} &= \partial_t\psi_{(2)} + im\Phi\psi_{(2)} - \frac{3im\Phi^2}{2}\psi_{(0)} - \Phi\partial_t\psi_{(0)} \,, \\
\mathcal{D}_i\psi_{(2)} &= \partial_i\psi_{(2)} \,,
\end{aligned} \tag{A.25}
$$

where, for convenience, we defined

$$\Phi = -\frac{GM}{r} \,. \tag{A.26}$$

This means that the NLO Lagrangian (3.86) becomes

$$
\begin{aligned}
\mathcal{L}_{\text{NLO}} = &-\sqrt{g}g^{ij}\left[\partial_i\psi^\star_{(0)}\partial_j\psi_{(2)} + \partial_i\psi^\star_{(2)}\partial_j\psi_{(0)}\right] - 2\sqrt{g}\Phi\partial_r\psi^\star_{(0)}\partial_r\psi_{(0)} \\
&+ \sqrt{g}\partial_t\psi_{(0)}\partial_t\psi^\star_{(0)} + 2im\sqrt{g}\Phi\left[\psi_{(0)}\partial_t\psi^\star_{(0)} - \psi^\star_{(0)}\partial_t\psi_{(0)}\right] \\
&+ im\sqrt{g}\left[\psi^\star_{(0)}\partial_t\psi_{(2)} - \psi_{(0)}\partial_t\psi^\star_{(2)} + \psi^\star_{(2)}\partial_t\psi_{(0)} - \psi_{(2)}\partial_t\psi^\star_{(0)}\right] \\
&- 2\sqrt{g}m^2\Phi\left[\psi_{(2)}\psi^\star_{(0)} + \psi^\star_{(2)}\psi_{(0)}\right] + 4\sqrt{g}m^2\Phi^2\psi_{(0)}\psi^\star_{(0)} \,,
\end{aligned} \tag{A.27}
$$

where $g_{ij}$ is the flat metric in spherical coordinates (cf., Section 3.3) satisfying $\sqrt{g} = r^2 \sin\theta$. Note that in Schwarzschild coordinates, we have that

$$\Phi + \frac{1}{2}h^{\rho\sigma}\Phi_{\rho\sigma} = 0 \,, \tag{A.28}$$

which means that the term involving the LO Lagrangian in (3.86) is zero. In other words, we have that

$$\tilde{\mathcal{L}}_{\text{NLO}} = \mathcal{L}_{\text{NLO}} \,, \tag{A.29}$$

in Schwarzschild coordinates.

On the other hand, in isotropic (spherical) coordinates, which we denote with a prime, the relevant derivatives are

$$\begin{aligned} \mathcal{D}_t \psi'_{(2)} &= \partial_t \psi'_{(2)} + im\Phi' \psi'_{(2)} - \frac{im}{2}\Phi'^2 \psi'_{(0)} - \Phi' \partial_t \psi'_{(0)} \,, \\ \mathcal{D}_{i'} \psi'_{(2)} &= \partial_{i'} \psi'_{(2)} \,, \end{aligned} \tag{A.30}$$

where

$$\Phi' = -\frac{GM}{r'} \,. \tag{A.31}$$

In contradistinction to what happens in Schwarzschild coordinates, the term involving the LO Lagrangian is no longer zero and instead depends on the number of spatial dimensions. In three spatial dimensions, we get

$$\Phi' + \frac{1}{2}h'^{\rho\sigma}\Phi'_{\rho\sigma} = -2\Phi' \,. \tag{A.32}$$

This means that the NLO Lagrangian now becomes

$$\begin{aligned} \mathcal{L}'_{\text{Iso,NLO}} &= -\sqrt{g'}g'^{ij}\left[\partial'_i \psi'^{\star}_{(0)}\partial'_j \psi'_{(2)} + \partial'_i \psi'^{\star}_{(2)}\partial'_j \psi'_{(0)}\right] - 2\sqrt{g'}\Phi' g'^{ij}\partial'_i \psi'^{\star}_{(0)}\partial'_j \psi'_{(0)} \\ &\quad + \sqrt{g'}\partial'_t \psi'_{(0)}\partial'_t \psi'^{\star}_{(0)} + 2im\sqrt{g'}\Phi'\left[\psi'_{(0)}\partial'_t \psi'^{\star}_{(0)} - \psi'^{\star}_{(0)}\partial'_t \psi'_{(0)}\right] + 2\sqrt{g'}m^2\Phi'^2 \psi'_{(0)}\psi'^{\star}_{(0)} \\ &\quad - 2\sqrt{g'}m^2\Phi'\left[\psi'_{(2)}\psi'^{\star}_{(0)} + \psi'^{\star}_{(2)}\psi'_{(0)}\right] + im\sqrt{g'}\left[\psi'^{\star}_{(0)}\partial'_t \psi'_{(2)} - \psi'_{(0)}\partial'_t \psi'^{\star}_{(2)}\right] \\ &\quad + i\sqrt{g'}m\left(\psi'^{\star}_{(2)}\partial'_t \psi'_{(0)} - \psi'_{(2)}\partial'_t \psi'^{\star}_{(0)}\right) - 2\Phi'\mathcal{L}'_{\text{LO}} \\ &= -\sqrt{g'}g'^{ij}\left[\partial'_i \psi'^{\star}_{(0)}\partial'_j \psi'_{(2)} + \partial'_i \psi'^{\star}_{(2)}\partial'_j \psi'_{(0)}\right] \\ &\quad + \sqrt{g'}\partial'_t \psi'_{(0)}\partial'_t \psi'^{\star}_{(0)} + 4im\sqrt{g'}\Phi'\left[\psi'_{(0)}\partial_t \psi'^{\star}_{(0)} - \psi'^{\star}_{(0)}\partial'_t \psi'_{(0)}\right] + 6\sqrt{g'}m^2\Phi'^2 \psi'_{(0)}\psi'^{\star}_{(0)} \\ &\quad - 2\sqrt{g'}m^2\Phi'\left[\psi'_{(2)}\psi'^{\star}_{(0)} + \psi'^{\star}_{(2)}\psi'_{(0)}\right] + im\sqrt{g'}\left[\psi'^{\star}_{(0)}\partial'_t \psi'_{(2)} - \psi'_{(0)}\partial'_t \psi'^{\star}_{(2)}\right] \\ &\quad + i\sqrt{g'}m\left(\psi'^{\star}_{(2)}\partial'_t \psi'_{(0)} - \psi'_{(2)}\partial'_t \psi'^{\star}_{(0)}\right) \,, \end{aligned} \tag{A.33}$$

where we used that

$$\mathcal{L}'_{\text{Iso,LO}} = im\sqrt{g'}\left[\psi'^{\star}_{(0)}\partial'_t \psi'_{(0)} - \psi'_{(0)}\partial'_t \psi'^{\star}_{(0)}\right] - 2\sqrt{g'}m^2\Phi'\psi'_{(0)}\psi'^{\star}_{(0)} - \sqrt{g'}g'^{ij}\partial'_i \psi'_{(0)}\partial'_j \psi'^{\star}_{(0)} \,. \tag{A.34}$$

We now want to explicitly change coordinates from Schwarzschild to isotropic. Using the relation between the radial directions in Schwarzschild and Isotropic coordinates (A.20), we find that

$$
\begin{aligned}
\Phi(r) &= \Phi'(r') + \frac{\Phi'^2(r')}{c^2} + \mathcal{O}(c^{-4}) \,, \\
\sqrt{g} &= \sqrt{g'} \left( 1 - \frac{2\Phi'(r')}{c^2} \right) + \mathcal{O}(c^{-4}) \,, \\
\partial_r &= \partial'_{r'} + \mathcal{O}(c^{-4}) \,.
\end{aligned}
\tag{A.35}
$$

The fact that both $\psi_{(0)}$ and $\psi_{(2)}$ transform as scalars under $c$-dependent coordinate transformations (A.23) implies that the only contributions to the NLO Lagrangian when changing coordinates from Schwarzschild to isotropic come from the LO Lagrangian (A.34). Explicitly, changing coordinates leads to

$$
\begin{aligned}
\mathcal{L}'_{\text{Schw,LO}} = \mathcal{L}'_{\text{Iso,LO}} + c^{-2} \big[ &- 2\mathrm{i}m\sqrt{g'}\Phi' \left( \psi'^{\star}_{(0)}\partial'_t\psi'_{(0)} - \psi'_{(0)}\partial'_t\psi'^{\star}_{(0)} \right) \\
&+ 2\sqrt{g'}m^2\Phi'^2\psi'_{(0)}\psi'^{\star}_{(0)} + 2\sqrt{g'}\Phi'\partial'_{r'}\psi'_{(0)}\partial'_{r'}\psi'^{\star}_{(0)} \big] \,,
\end{aligned}
\tag{A.36}
$$

where the terms at order $c^{-2}$ will contribute to the NLO Lagrangian: adding these terms to $\mathcal{L}'_{\text{Schw,NLO}}$ precisely leads to $\mathcal{L}'_{\text{Iso,NLO}}$ in (A.33).

Now we turn our attention to $\hat{\mathcal{L}}_{\text{Schw,NLO}}$ and $\hat{\mathcal{L}}'_{\text{Iso,NLO}}$, which are expressed in terms of the normalised wave functions (A.24). In Schwarzschild coordinates, the operator $\hat{O}_{\text{Schw}}$ is given by

$$
\hat{O}_{\text{Schw}} = \frac{\mathrm{i}}{2m}\partial_t - \Phi \,,
\tag{A.37}
$$

and hence the NLO Lagrangian (A.27) can be expressed in terms of $\hat{\psi}_{\text{Schw}(2)}$ as

$$
\begin{aligned}
\hat{\mathcal{L}}_{\text{Schw,NLO}} = &-\sqrt{g}g^{ij} \left[ \partial_i\psi^{\star}_{(0)}\partial_j\hat{\psi}_{\text{Schw}(2)} + \partial_i\hat{\psi}^{\star}_{\text{Schw}(2)}\partial_j\psi_{(0)} \right] - 2\sqrt{g}\Phi\partial_r\psi^{\star}_{(0)}\partial_r\psi_{(0)} \\
&- \sqrt{g}\partial_t\psi_{(0)}\partial_t\psi^{\star}_{(0)} - \sqrt{g}g^{ij} \left( 2\Phi\partial_i\psi_{(0)}\partial_j\psi^{\star}_{(0)} - \frac{\mathrm{i}}{m}\partial_i\psi^{\star}_{(0)}\partial_j\partial_t\psi_{(0)} \right) \\
&- \sqrt{g}\partial_r\Phi(\psi_{(0)}\partial_r\psi^{\star}_{(0)} + \psi^{\star}_{(0)}\partial_r\psi_{(0)}) - 2\sqrt{g}m^2\Phi \left[ \hat{\psi}_{\text{Schw}(2)}\psi^{\star}_{(0)} + \hat{\psi}^{\star}_{\text{Schw}(2)}\psi_{(0)} \right] \\
&+ \mathrm{i}m\sqrt{g} \left[ \psi^{\star}_{(0)}\partial_t\hat{\psi}_{\text{Schw}(2)} - \psi_{(0)}\partial_t\hat{\psi}^{\star}_{\text{Schw}(2)} + \hat{\psi}^{\star}_{\text{Schw}(2)}\partial_t\psi_{(0)} - \hat{\psi}_{\text{Schw}(2)}\partial_t\psi^{\star}_{(0)} \right] \,.
\end{aligned}
\tag{A.38}
$$

In isotropic coordinates, the expression for the operator $\hat{O}'_{\text{Iso}}$ is *different*:

$$
\hat{O}'_{\text{Iso}} = \frac{\mathrm{i}}{2m}\partial'_t - 2\Phi' \,.
\tag{A.39}
$$

The origin of this is, as we alluded to above, the fact that the normalised wave functions transform in a non-standard way under $c$-dependent coordinate transformations. To see why, consider the inner product (3.52) that allowed us to read off

the operator $\hat{O}$. In Schwarzschild coordinates, we can write this inner product as

$$\langle\varphi_{\text{KG}}|\psi_{\text{KG}}\rangle = 2m\int_{\Sigma}d^3x\sqrt{g}\left[\psi_{(0)}\varphi_{(0)}^{\star} + c^{-2}\psi_{(0)}(\hat{O}_{\text{Schw}}\varphi_{(2)})^{\star} + c^{-2}\varphi_{(0)}^{\star}\hat{O}_{\text{Schw}}\psi_{(2)} + \cdots\right],$$
(A.40)

where $\hat{O}_{\text{Schw}}$ is defined in (A.37), and $\Sigma$ is a constant-time hypersurface. Changing coordinates from Schwarzschild to isotropic, we get an extra contribution at order $c^{-2}$

$$\langle\varphi'_{\text{KG}}|\psi'_{\text{KG}}\rangle = 2m\int_{t=\text{cst}}d^3x'\sqrt{g'}\left[\psi'_{(0)}\varphi'^{\star}_{(0)} + c^{-2}\psi'_{(0)}(\hat{O}'_{\text{Schw}}\varphi'_{(2)} - \Phi'\varphi'_{(0)})^{\star}\right.$$
$$\left. + c^{-2}\varphi'^{\star}_{(0)}\left(\hat{O}'_{\text{Schw}}\psi'_{(2)} - \Phi'\psi'_{(0)}\right) + \cdots\right],$$
(A.41)

which correctly reproduces the following relation between (A.37) and (A.39)

$$\hat{O}'_{\text{Iso}}\psi'_{(2)} = \hat{O}'_{\text{Schw}}\psi'_{(2)} - \Phi'\psi'_{(0)}.$$
(A.42)

Using that the un-normalised wave function $\Psi$ transforms as a scalar (A.23), we get the relation

$$\hat{\Psi}'_{\text{Schw}} = \hat{\Psi}'_{\text{Iso}} + \frac{1}{c^2}\Phi'\hat{\Psi}'_{\text{Iso}} + \mathcal{O}(c^{-4}),$$
(A.43)

where

$$\hat{\Psi}'_{\text{Schw}} = \psi'_{(0)} + \psi'_{\text{Schw}(2)}\qquad\text{and}\qquad\hat{\Psi}'_{\text{Iso}} = \psi'_{(0)} + \psi'_{\text{Iso}(2)},$$
(A.44)

respectively. To get the NLO Lagrangian in istropic coordinates, we must add to (A.38) the terms that arise from the LO Lagrangian, given again by (A.36), and also express the Schwarzschild wave function in terms of the isotropic wave function as per (A.43).

### A.2.3 Schrödinger's equation

Setting $J = 0$ in the equation for Kerr (4.20) gives rise to the following Schrödinger equation in Schwarzschild coordinates up to order $c^{-2}$

$$i\partial_t\hat{\Psi}_{\text{Schw}} = -\frac{1}{2m}\Delta\hat{\Psi}_{\text{Schw}} - \frac{GmM}{r}\hat{\Psi}_{\text{Schw}} + \frac{GM}{c^2rm}\partial_r^2\hat{\Psi}_{\text{Schw}} - \frac{mG^2M^2}{2c^2r^2}\hat{\Psi}_{\text{Schw}}$$
$$+ \frac{GM}{2mc^2r}\Delta\hat{\Psi}_{\text{Schw}} - \frac{1}{8c^2m^3}\Delta^2\hat{\Psi}_{\text{Schw}} + \frac{GM}{4mc^2}\Delta(r^{-1})\hat{\Psi}_{\text{Schw}}$$
$$+ \frac{GM}{2mc^2r^2}\partial_r\hat{\Psi}_{\text{Schw}}.$$
(A.45)

We have the following useful relations between various quantities in Schwarzschild and isotropic coordinates

$$\frac{1}{r} = \frac{1}{r'} - \frac{GM}{c^2 r'^2} + \mathcal{O}(c^{-4}),$$

$$\frac{1}{r^2} = \frac{1}{r'^2} - \frac{2GM}{c^2 r'^3} + \mathcal{O}(c^{-4}),$$

$$\partial_r = \left(\frac{dr}{dr'}\right)^{-1} \partial_{r'} = \frac{r'^2}{r'^2 - \frac{G^2 M^2}{4c^4}} \partial_{r'} = \partial_{r'} + \mathcal{O}(c^{-4}),$$

$$\Delta f = \Delta' f' - \frac{2GM}{r'c^2} \Delta' f' + \frac{2GM}{c^2 r'^2} \partial_{r'} f' + \frac{2GM}{c^2 r'} \partial_{r'}^2 f',$$

(A.46)

where $f(t, r, \theta, \phi) = f'(t, r', \theta, \phi)$. Combining these with the relation (A.43) we get

$$i\partial_t \hat{\Psi}'_{\text{Iso}} = -\frac{1}{2m} \Delta' \hat{\Psi}'_{\text{Iso}} + m\Phi' \hat{\Psi}'_{\text{Iso}} - \frac{1}{8c^2 m^3} \Delta'^2 \hat{\Psi}'_{\text{Iso}} + \frac{m\Phi'^2}{2c^2} \hat{\Psi}'_{\text{Iso}} - \frac{3\Delta'\Phi'}{4mc^2} \hat{\Psi}'_{\text{Iso}}$$
$$- \frac{3\Phi'}{2mc^2} \Delta' \hat{\Psi}'_{\text{Iso}} - \frac{3}{2mc^2} \partial_{r'}\Phi' \partial_{r'} \hat{\Psi}'_{\text{Iso}},$$

(A.47)

and thus turns the Schrödinger equation in Schwarzschild coordinates (A.45) into the Schrödinger equation in isotropic coordinates we obtained in (A.17).

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

Backgrounds and Symmetries of the Lifshitz Vacuum," JHEP **08** (2015) 006,
arXiv:1502.00228 [hep-th].

[56] J. B. Hartle and K. S. Thorne, "Slowly Rotating Relativistic Stars. II. Models for
Neutron Stars and Supermassive Stars," Astrophys. J. **153** (1968) 807.

[57] B. Mashhoon, "Neutron interferometry in a rotating frame of reference," Phys. Rev.
Lett. **61** (1988) 2639–2642.

[58] B. Mashhoon, "On the coupling of intrinsic spin with the rotation of the earth,"
Physics Letters A **198** no. 1, (1995) 9–13.
https://www.sciencedirect.com/science/article/pii/037596019500010Z.

[59] A. J. Silenko, "Quantum-mechanical description of Lense-Thirring effect for
relativistic scalar particles," Phys. Part. Nucl. Lett. **10** (2013) 637–641,
arXiv:1408.2226 [gr-qc].

[60] A. S. Eddington, The mathematical theory of relativity. The University Press, 1923.