# Peer review of "A coupling prescription for post-Newtonian corrections in Quantum Mechanics"

_SciPost Physics_

## Round 1 · Referee Report · Efe Hamamci (Referee 1) · 2023-11-26

Strengths

1 - The paper is well structured, easy to follow and technical details are adequately provided. 2 - Symmetries of the background metric are thoroughly discussed in section 2. 3 - The coupling prescription is explicitly demonstrated in section 3- The results of section 4 are in accord with the ones found in the literature while paving way for possible directions. 4- Demonstrates the interplay between quantum theory and gravity very well.

Report

The paper begins with the development of nonrelativistic gravity through expansion of Lorentzian geometry in inverse powers of speed of light $(1/c^2)$ with high emphasis on symmetries. The resultant background metric is flat at leading order with post-Newtonian corrections coming at next-to-leading order. Upon deriving the flat Schrödinger equation and its subleading $1/c^2$ correction using a WKB-like ansatz and $1/c^2$ expansion on flat KG equation, the equations are coupled to the background geometry via a symmetry based prescription. The standard $L^2(\mathbb R^3)$ inner product of the resulting theory is obtained via $1/c^2$ expansion on the KG inner product involves a background dependent redefinition of the subleading Schrödinger field. The uniquely fixed form of the coupled equations are also verified through a Lagrangian established by $1/c^2$ expansion on minimally coupled complex KG Lagrangian. Finally, the expansion formalism is applied to the Kerr geometry resulting in a generalised form of the Lense-Thirring metric leading to a novel Hamiltonian capturing an possibly observable rotational effect.

The paper is well structured, easy to follow and technical details are adequately provided. The interplay between quantum theory and weak gravity is successfully established while also paving way for possible future prospects. Specifically, symmetries of the background metric are thoroughly discussed in section 2 and in section 3 the coupling prescription is applied in clear steps. The results of section 4 are in accord with the ones found in the literature while providing experimentally observable results. In our point of view, the paper is almost suitable for publication in SciPost as is. However, we kindly ask the authors to address several minor typos before publication:

1- In eqn. (3.50) the field $\Phi$ is used for KG field whereas the same letter was used in eqn. (2.11) for the metric field. 2- Repeated lower indices in eqns. (3.69) and (3.70) 3- $L$ appears inside eqn. (3.76) 4- $\psi$ appears twice above eqn. (4.24)

  • validity: -
  • significance: -
  • originality: -
  • clarity: -
  • formatting: -
  • grammar: -

Author:  Emil Have  on 2024-02-03  [id 4299]

(in reply to Report 1 by Efe Hamamci on 2023-11-26)
Category:
answer to question
correction

We are grateful to the referee for their time and their positive comments. We address the comments in the attached document.

Attachment:

Response_1.pdf

---

## Round 1 · Referee Report · Philip Schwartz (Referee 2) · 2023-12-19

Report

In the manuscript ‘A coupling prescription for post-Newtonian corrections in Quantum Mechanics’, the authors discuss the description of quantum systems under post-Newtonian gravity in a geometric post-Newton–Cartan formalism. They approach this problem from two different angles: first, they develop a symmetry-based ‘minimal coupling’ scheme to couple the $1/c$ expansion of the Minkowski spacetime Klein–Gordon equation to first-order post-Newton–Cartan gravity (for flat leading-order geometry); second, they show that the same result arises from expanding the Klein–Gordon action in $1/c$ and varying the resulting action. The authors exemplify their formalism by applying it on a Kerr background.
The manuscript is well-written, and its subject is of very high interest to researchers working at the interface of quantum mechanics and gravity. I have not checked all computations in detail, but those I did check were correct; all other results are plausible and their derivations are provided with many details. I believe the manuscript to be highly relevant: systematic (low-energy approximative) descriptions of quantum mechanics in weak gravitational backgrounds based on post-Newtonian expansions of ‘relativistic’ wave equations have been discussed in the literature for quite some time, but this article is the first one to use a fully geometric formalism for this endeavour. By doing so, the authors pay full tribute to gravity being an inherently geometric phenomenon.
I am sure that this paper will yield much follow-up work, it having paved the way for discussing post-Newtonian quantum mechanics from a ‘covariant’, truly geometric perspective, thus contributing an essential ingredient to the very active current research on the interface of (low-energy) quantum mechanics and gravity. Hence, it is certainly a very good fit for SciPost Physics. However, I have some points that I ask the authors to address before publication.
(Note that even though the following list is quite long, this is not in any way due to the paper needing a lot of changes, but rather due to me being very detailed. I genuinely believe this to be a truly great paper.)

Requested changes

  1. In section 3.3, the authors intend to show how to rewrite the equation obtained previously in (spatial) Cartesian coordinates in spherical coordinates. However, I believe that this section may (and should!) in a very easy way be simultaneously simplified, shortened and vastly generalised, by the following observation: the rewriting in this section completely amounts to a standard rewriting of the equations obtained previously in Cartesian coordinates in a differential-geometric way that is independent of the choice of (spatial) coordinates. Concretely, this means recognising spatial partial derivatives in Cartesian coordinates as covariant derivatives with respect to the Levi-Civita connection $\nabla$ of the induced spatial metric $h_{ij}$, expressed in Cartesian coordinates, such that expressions such as $\delta^{ij} \partial_i \partial_j = h^{ij} \nabla_i \nabla_j$ may directly be expressed in coordinate-invariant form. In particular, this form of the equations then applies in any spatial coordinate system—nothing depends on the coordinates being spherical coordinates. I strongly suggest the authors to implement this. Also, I believe that the standard notational practice (standard in differential geometry and relativity) of denoting geometric objects (such as $\psi$) by the same symbol, independently of which coordinates are used, should be followed, i.e. that the primes on $\psi'$ etc. should be dropped. The same applies to the very end of section 3.4 (top of p. 26): this should also be called the form of the equations in arbitrary spatial coordinates (not necessarily spherical ones), and the primes on the symbols should be dropped.

  2. Regarding appendix A.2, in which the form of the equations in an expanded Schwarzschild background in different coordinates is compared, I want to remark a few things. First, the phrasing should be corrected in that $\Psi$ transforms as a scalar under spatial coordinate transformations (the manuscript speaks of ‘coordinate transformations’ in general, but as soon as time is involved, $\Psi$ will change). Second, the derivations can also be clarified and somewhat shortened by differential geometric argumentation: in general, the rescaled wave functions $\hat\Psi, \hat\Phi$ in terms of which the scalar product takes the standard $\mathrm{L}^2$ form are not scalar functions on the spatial hypersurface $\Sigma$, but scalar densities of weight $1/2$, such that the integral $\int_\Sigma d^3x \hat\Psi^* \hat\Phi$ is well-defined (i.e. takes the same form in any spatial coordinate system). On the one hand, this means that they will transform differently to scalar functions not only under $c$-dependent coordinate transformations, but also under $c$-independent ones (if these change to non-Cartesian coordinates). On the other hand, it also allows us to directly compute their transformation behaviour under the transformation from Schwarzschild to isotropic coordinates, without any need to explicitly compute the expansion of the KG inner product in both coordinate systems (which is done in the manuscript after (A.24)): by applying the general transformation behaviour of scalar densities, which reads $\hat\Psi(x) \to \hat\Psi'(x') = (\det(\partial x^i/\partial x'^j))^{-1/2} \hat\Psi(x(x'))$, to the particular transformation from Schwarzschild to Isotropic coordinates, one directly obtains (A.43). Concretely, in the Schwarzschild → isotropic case at hand, we have

    $$\hat\Psi_\text{Schw}(r',\theta,\phi) = \left(\frac{r^2}{r'^2} \frac{\partial r}{\partial r'}\right)^{-1/2} \hat\Psi_\text{Iso}(r',\theta,\phi) ,$$
    and computing the prefactor to the necessary order yields $\left(\frac{r^2}{r'^2} \frac{\partial r}{\partial r'}\right)^{-1/2} = 1 - \frac{GM}{c^2 r'}$, which gives the transformation behaviour from (A.43).

  3. I have some minor wording/naming suggestions:

  4. The authors call the redefined wave functions in terms of which the post-Newtonian quantum theory’s scalar product takes the standard $\mathrm{L}^2$ form ‘normalised wave functions’. This is somewhat misleading, since ‘normalised’ can easily be misinterpreted as meaning ‘of norm 1’. I suggest to use some other name, such as ‘flat-representation wave function’ or something similar. (I don’t think that suggestion is particularly good, but to me it’s more clear than ‘normalised’.)
  5. After (3.84), $K_{\mu\nu}$ is called ‘extrinsic curvature’, which is misleading. It is true that this is the leading order term in the $1/c$ expansion of the extrinsic curvature of the spatial hypersurface in Lorentzian geometry; but in Galilei geometry, there is no sensible (invariant) notion of extrinsic curvature. I think that $K_{\mu\nu}$ should be either given no name or described as the leading order term of the extrinsic curvature’s expansion. I also suggest to mention the invariant geometric interpretation of $K = h^{\mu\nu} K_{\mu\nu}$, namely that it is the expansion (in the congruence sense) $K = \nabla_\mu v^\mu$ of $v$ with respect to any torsion-free Galilei connection $\nabla$ on $(M,\tau,h)$.

  6. The paper [24] doesn’t use a WKB-like ansatz for a solution to a wave equation in order to derive post-Newtonian corrections; thus it should not be cited in the corresponding places at the top of p. 5 and directly before (3.1). However, the paper [51] should already be cited at the top of p. 5 (since it was, to my knowledge, the first since [30] that explicitly developed the method further). Also, both [42] and [52] should already be cited in the middle of p. 2 together with [22-26], since they also deal with the interface of low-energy quantum systems with gravity.

  7. At a few points, I found what I believe to be minor errors, which should be corrected:

  8. Before (2.3), the formulation ‘their inverses’ is somewhat misleading, since these objects are not inverses in the strict sense.
  9. The last sentence on p. 8 states ‘A Galilean structure is entirely determined by the properties of the clock form’. In the present formulation, this is false, since the space metric $h$ is also part of the Galilei structure (and clearly not determined by $\tau$). I suggest to reformulate this to ‘the causal structure’ or the like (which seems to be what the authors intended to say).
  10. In the caption of Figure 1, the formulation of $v$ being ‘orthogonal to the equal-time hypersurfaces’ doesn’t make geometric sense in the absence of a non-degenerate metric—the authors probably mean ‘complementary’ instead of ‘orthogonal’. (Also, in my opinion $v$ being ‘observer-dependent’ is not really worth a comment—it’s a four-velocity, after all!)
  11. The decomposition (2.12) assumes $X^{\mu\nu}$ to be a symmetric contravariant 2-tensor, so this should be added before (2.12). I also suggest to mention that (2.12) is the decomposition of $X$ into space- and timelike parts w.r.t. $v$.
  12. Before (2.19), the generator $\Xi$ should be spoken of as a ‘vector field’, not a ‘vector’.
  13. At the beginning of section 3.4, taking for the quantum theory constructed from symmetry considerations the scalar product that arises from the (curved) KG inner product amounts to a definition and cannot be ‘shown’ (in the sense of ‘proved’), so this phrasing should be changed. (Of course, later when the same theory is constructed from expanding the KG action, this scalar product naturally arises.)

  14. The following references should be added:

  15. For the formal expansion of Lorentzian geometry, extending Newton–Cartan gravity (p. 4): G. Dautcourt, CQG 14, A109 (1997), arXiv:gr-qc/9610036; and W. Tichy, É. É. Flanagan, PRD 84, 044038 (2011), arXiv:1101.0588
  16. When referring to MTW’s chapter 12 for a pedagogical introduction to Newton–Cartan gravity at the beginning of sec. 2, I suggest to also refer to chapter 4 of the following book: David B. Malament, Topics in the Foundations of General Relativity and Newtonian Gravitation Theory, University of Chicago Press (2012)
  17. In sec. 3.5, the leading-order Newton–Cartan Schrödinger theory in (3.81), (3.82) was already constructed by Duval and Künzle in 1984, which certainly needs to be acknowledged: C. Duval, H. P. Künzle, Gen. Relativ. Gravit. 16 (1984), 333–347, DOI 10.1007/BF00762191

  18. The following are some rather small changes/clarifications that I ask the authors to incorporate:

  19. In the introduction's very first sentence, it should be made clear that ‘all tests so far’ means tests of GR proper, since Newtonian gravity and the weak equivalence principle have been tested with quantum systems.
  20. At the bottom of p. 2, where minimal coupling is described, ‘derivatives’ should be replaced by ‘partial derivatives in inertial coordinates’, since this is more clear.
  21. Before (1.1), I suggest to explicitly speak of ‘the wave function $\Psi$ of a point particle’.
  22. In the last par. on p. 3, I suggest that ‘gravitational forces’ be replaced by ‘gravitational effects’—the basic lesson of GR is that gravity is not a force!
  23. In the 2nd full par. on p. 5, I suggest to not use implicit summation over i, but explicitly write $h^{\mu\nu} = \delta^\mu_i \delta^\nu_j \delta^{ij}$.
  24. At the end of the introduction, when discussing the structure of the paper, when describing sec. 2.2 I suggest to replace ‘where we also consider a flat Galilean structure’ by ‘where we also discuss flat Galilean structures’.
  25. In the first sentence of sec. 2, I suggest to replace ‘is obtained by expanding’ by ‘may be obtained by expanding’, since a $1/c$ expansion is not the only way of implementing the passage from Poincaré- to Galilei-invariant physics. Related to that, at the beginning of section 3, before (3.7) the formulation ‘As we discussed above, the inclusion of $1/c^2$ corrections implies that’ should be replaced by ‘As we discussed above, to derive $1/c^2$ corrections, we assume that’.
  26. Before (2.6), it should be made explicit that $\tau \wedge d\tau = 0$ is implied by the Einstein equations for suitable matter (i.e. for an energy–momentum tensor with suitable $1/c$ expansion behaviour).
  27. Before (3.8), I suggest to explicitly state that this is transformation as a scalar.
  28. Around (3.11), the time translation transformation should be explained more clearly: I suggest to make explicit that we define $\Psi'$ by writing $\phi_\text{KG}(t' - t_0,x) = \phi_\text{KG}(t,x) =: \mathrm{e}^{-imc^2 t'} \Psi'(t',x')$, and to use $t'$ instead of $t$ as argument in (3.11) (since this is the form of the equation that one obtains from this definition).
  29. After (3.35), I suggest to use a different name than ‘field strength’ for $M$, since $m$ is not really a gauge potential—perhaps simply say ‘exterior derivative’?
  30. Directly before (4.11), I don’t understand the formulation ‘Making factors of $c$ explicit in (4.4)’, since they already are explicit. What is meant here, simply insertion of (4.5)?
  31. In the 3rd sentence of the discussion/outlook section, the authors claim that they could ‘use symmetry to uniquely fix the form of the equations’. This is not entirely true, since only some ‘minimal’ form of the equations is fixed. I suggest to use a formulation along the lines of ‘use symmetry to uniquely fix the form of the equations up to “non-minimal” terms’.

  32. Some additions I suggest the authors to make:

  33. At the end of section 3.2, one could perhaps add a comment that $\mathcal D_\mu$ acts according to an expansion of the Poincaré algebra. (That this would be nice to better understand is also mentioned in the discussion/outlook—In my opinion it also deserves to be mentioned already here!)
  34. Before (3.54), I suggest to explicitly write ‘to take the standard $\mathrm{L}^2(\mathbb{R}^d)$ form’.
  35. In eq. (3.83), the RHS may be rewritten as $-\frac{1}{2m} h^{\mu\nu} \mathcal D_\mu \mathcal D_\nu \psi_{(0)}$, which is clearer to read.
  36. Before (4.25), I suggest to emphasise that we know that some terms need to ‘conspire’: we know that the KG inner product is conserved in the full KG theory, so we know that the $1/c$-expanded Hamiltonian has to be Hermitian!

  37. A few typos I found:

  38. In the last full par. on p. 4, ‘local Lorentz and diffeomorphisms’ should read ‘local Lorentz transformations and diffeomorphisms’.
  39. On p. 6, the last sentence of the top paragraph is incomplete.
  40. The first two sentences of the paragraph before (4.13) (from ‘If we assume’ to ‘in Section A.2.’) should probably be appended to the previous paragraph, since they still explain (4.11).
  41. At the bottom of p. 31, ‘LO Schrödinger equations’ should read ‘LO Schrödinger equation’.
  42. (4.22) should have $\hat\Phi$ and $\hat\Psi$ instead of $\phi$ and $\psi$.
  43. After (4.23), ‘for the fields $\psi,\psi$’ should read ‘for the fields $\psi,\phi$’.
  44. In the 4th sentence of the discussion/outlook section, ‘Complimentary to a “bottom-up” perspective’ should read ‘Complementary to this “bottom-up” perspective’.
  45. At the end of the 2nd / the beginning of the 3rd paragraph of the outlook section, there is a double mention of extending the formalism to include electromagnetism and spin.
  46. In the title of section A.2.2, ‘Schrödinger's Lagrangian’ should read ‘Schrödinger Lagrangian’.

  47. Some minor suggestions (that I would appreciate, but with which not being implemented I am also fine):

  48. At the very top of p. 2, the paper arXiv:2310.03719 might be cited, since it is a recent extension of [8]. (Disclaimer: I am a co-author of this paper.)
  49. When [17,18] are cited, some more recent papers on the qBOUNCE experiment might be cited, e.g. arXiv:2301.05984, arXiv:2301.08583.
  50. In the 3rd par. on p. 2, I suggest to replace ‘that arise from fixed GR backgrounds’ by ‘that arise from fixed gravitational backgrounds’, since such a background might arise from different metric theories of gravity.
  51. Personally, I do not really like the terminology ‘nonrelativistic’, since so-called ‘nonrelativistic’ physics of course also satisfies a (perhaps locally realised) relativity principle, just a Galilean instead of a Lorentzian one. Therefore, I personally would prefer to rephrase passages using the term ‘nonrelativistic’ by saying something like ‘Newtonian’ or ‘Galilean’ instead, e.g. ‘formal post-Newtonian expansion’ or the like instead of ‘nonrelativistic expansion’. However, I want to make explicit again that this is only a suggestion of mine, with which not being implemented I am also fine.

  • validity: top
  • significance: high
  • originality: high
  • clarity: high
  • formatting: excellent
  • grammar: excellent

Author:  Emil Have  on 2024-02-03  [id 4300]

(in reply to Report 2 by Philip Schwartz on 2023-12-19)

We are grateful to the referee for their very careful reading of our manuscript and for the many useful suggestions, which have improved the quality of the paper. We address the comments in the attached document.

Attachment:

Response_2.pdf

---

## Editorial Decision

resubmitted